# Towards Near-Real-Time Telemetry-Aware Routing with Neural Routing Algorithms

## Abstract

Routing algorithms are crucial for efficient computer network operations, and in many settings they must be able to react to traffic bursts within milliseconds. Live telemetry data can provide informative signals to routing algorithms, and recent work has trained neural networks to exploit such signals for traffic-aware routing. Yet, aggregating network-wide information is subject to communication delays, and existing neural approaches either assume unrealistic delay-free global states, or restrict routers to purely local telemetry. This leaves their deployability in real-world environments unclear. We cast telemetry-aware routing as a delay-aware closed-loop control problem and introduce a framework that trains and evaluates neural routing algorithms, while explicitly modeling communication and inference delays. On top of this framework, we propose LOGGIA, a scalable graph neural routing algorithm that predicts log-space link weights from attributed topology-and-telemetry graphs. It utilizes a data-driven pre-training stage, followed by on-policy Reinforcement Learning. Across synthetic and real network topologies, and unseen mixed TCP/UDP traffic sequences, LOGGIA consistently outperforms shortest-path baselines, whereas neural baselines fail once realistic delays are enforced. Our experiments further suggest that neural routing algorithms like LOGGIA perform best when deployed fully locally, i.e., observing network states and inferring actions at every router individually, as opposed to centralized decision making.

## 1 Introduction

Routing denotes the task of finding paths for data packets to travel from senders to receivers. It enables and optimizes data transfer and is thus an essential component of any computer network. Conventional routing protocols like Open Shortest-Path First (OSPF) (Moy, 1997), Enhanced Interior Gateway Routing Protocol (EIGRP) (Savage et al., 2016) and Routing Information Protocol (RIP) (Hedrick, 1988) obtain paths, e.g., via Dijkstra's algorithm with link costs calculated from the network's topology, or via the Bellman–Ford algorithm with hop-count metrics. Optimizing routing has been a long-standing subject of research (Xiao et al., 2021) due to unpredictable traffic patterns (Alizadeh et al., 2014; Wendell & Freedman, 2011) or link/node failures that change the underlying network topology (Gill et al., 2011; Markopoulou et al., 2008; Turner et al., 2010). Conventional algorithms often react too slowly to such events, letting outdated or broken routing paths cause drops in service quality (Benson et al., 2011). In an increasing number of networking setups, traffic dynamics and topology changes may require optimizing and deploying new routing configurations within milliseconds (Gay et al., 2017a), and leveraging telemetry data for the adjustment to new situations (Turkovic et al., 2018; Xie et al., 2022). This fine-grained telemetry-aware variant of routing can be viewed as a closed-loop control problem with a millisecond control frequency, as illustrated in Figure 1.

Providing quick routing optimization from high-dimensional network states is challenging for conventional routing algorithms, not least due to the exponentially increasing solution space (Xu et al., 2011). Therefore, leveraging Machine Learning (ML) for such algorithms has been a popular and long-standing subject of research (Boyan & Littman, 1993; Xiao et al., 2021). Recent work has included telemetry data into their closed-loop routing algorithms (Bernárdez et al., 2023; Boltres et al., 2024; Gui et al., 2024), but ignores the networked nature of routing in computer networks. The works of Bernárdez et al. (2023) and Boltres et al.

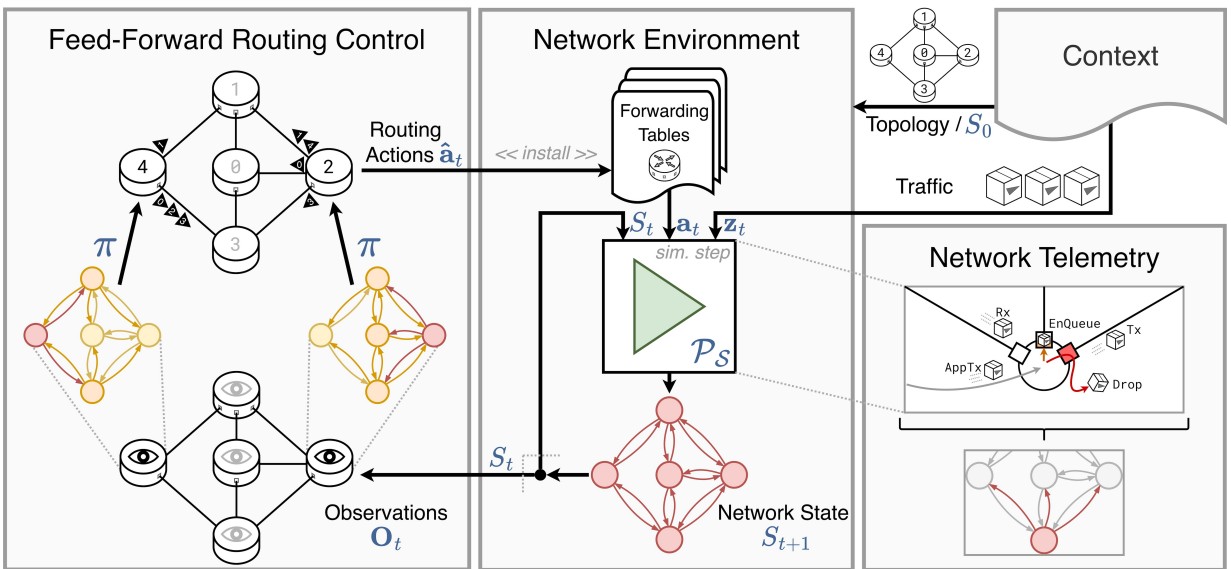

Figure 1: We view telemetry-aware near-real time routing as a closed-loop control problem, illustrated here with distributed *observers* and *routing agents*. The network environment (center) operates on an externally provided network topology (upper right) and evolves a network state $S_t$, given populated forwarding tables $\mathbf{a}_t$ and upcoming traffic arrivals $\mathbf{z}_t$. During operation, telemetry information is used to form local network snapshots (lower right, shown for node 3 and incident links). The new state $S_{t+1}$, is the union of all local snapshots, and observers obtain localized views $\mathbf{O}_t$ thereof (lower left, shown for nodes 2 and 4). Routing policies $\pi$, which can include neural networks or conventional algorithms, calculate new routing decisions $\hat{\mathbf{a}}_t$ from the provided observations (upper left), which are then used to update the forwarding table.

(2024) present centralized routing algorithms that disregard delays incurred by e.g. propagating state and control information. On the other hand, the model proposed in Gui et al. (2024) distributes routing control to the routers, which however rely on purely local telemetry information to make their decisions. It appears to be an open question whether neural routing algorithms converge to good routing policies using real-time telemetry data, while respecting communication delays at the same time. Yet, delay-aware algorithms are necessary to deploy neural routing algorithms in real-world networking scenarios.

In this work, we take a step towards deploying neural routing algorithms in real-world computer networks, by extending the capabilities of existing approaches and the network models they are trained in. Concretely, this paper includes the following contributions: **(i)** We present a packet-level network simulation framework capable of training and evaluating centralized or decentralized neural routing algorithms (Section 3), while respecting communication and neural network inference delays. **(ii)** We propose our new neural routing algorithm $LOGGIA$[1], which predicts link weights in logarithmic space for subsequent least-cost paths routing (Section 4). Our experiments show that $LOGGIA$ delivers more data than conventional routing protocols and neural baseline algorithms in our more realistic delay-aware routing setting (Sections 5 and 6), scaling to large and unseen networks even when trained on just a single network topology. **(iii)** We demonstrate the importance of respecting communication and inference delays in the network model. Our results suggest to fully distribute state observation and routing agent modules during deployment, i.e., let the individual routing nodes of a network observe the state and derive routing actions locally, whereas training converges well enough with a centralized state observer.

## 2   Related Work

**ML for Routing Algorithms.** Utilizing ML for network routing algorithms has been a subject of study for decades (Boyan & Littman, 1993; Choi & Yeung, 1995). From these early approaches using tabular

---

[1] $LOGGIA$ = "LOg-space link weight prediction on Graphs with Guided update epochs and Implicit-Alpha entropy adaptation".

Reinforcement Learning (RL), recent research has evolved to consider various problem formulations, network types, use cases and goals. We refer to the surveys in Xiao et al. (2021) and Jiang et al. (2024) for a more extensive overview. A large subcategory considers routing optimization as a component of Traffic Engineering (TE) which, as stated in Farrel (2024) for internet traffic, includes "[...] the measurement, characterization, modeling and control [...]" of the network and its traffic. While the official definition also encompasses sub-second control actions, e.g., for packet routing, ML-powered research geared at TE generally derives splitting ratios for precomputed sets of shortest paths (Xiao et al., 2021; Guo et al., 2022; Rusek et al., 2022; Bernárdez et al., 2023; Xu et al., 2023; Gui et al., 2024; Guo et al., 2025) from traffic aggregated over minutes or hours of operation. A notable exception is *RedTE* (Gui et al., 2024), which distributes routing control and measures traffic in milliseconds. Yet, its distributed routing modules only work with their own respective traffic information and do not communicate with each other, and they require to be trained anew for each topology. Aforementioned ML-based approaches for TE aim to solve a multi-commodity network flow problem (Ahuja et al., 1994) and, more generally, a combinatorial optimization problem (Papadimitriou & Steiglitz, 1998). For problems adjacent to computer network routing, recent research has explored RL algorithms (Khalil et al., 2017; Nazari et al., 2018; Kool et al., 2019), too, that however focus predominantly on a sequential decoding of a learned embedding of the input graph. In general, exploring the relation between neural networks operating on graphs (so-called Graph Neural Networks (GNNs)) and conventional algorithms for combinatorial optimization problems appears to be an active area of research (Dudzik & Veličković, 2022). A different line of work on ML-powered computer network routing operates on a finer temporal scale and infers routes for individual packets (Boyan & Littman, 1993; Xiao et al., 2020; Mai et al., 2021). These approaches do not scale to modern large-scale networks, in which routers need to forward individual packets at line speed, which may be as quick as a few nanoseconds per packet. In contrast, our approach modifies forwarding tables on a millisecond scale, thereby achieving near-real time control capabilities. Other approaches take routing decisions on the level of flows or per destination node (Bhavanasi et al., 2022; Boltres et al., 2024). However, they either do not support telemetry features (Bhavanasi et al., 2022), or they train a centralized controller using "birds-eye" network states that disregard communication delays (Boltres et al., 2024). Our approach mitigates these two limitations, supporting attributed state graphs with arbitrary telemetry features, as well as delay-aware routing control.

**Data and Environments for Telemetry-Aware Neural Routing.** Telemetry-aware routing is a sequential decision problem, and training neural algorithms for this task requires sufficiently general training data covering the varying network settings and use cases. Unlike, e.g., for computer vision (Deng et al., 2009) or text processing (Gao et al., 2020), there are no large-scale datasets available for training neural routing algorithms. Available data includes collections of network topologies (Spring et al., 2002; Knight et al., 2011) or traffic traces (CAIDA), and small-scale datasets that combine topology and traffic data (Mestres et al., 2017), but not the combination of topology, temporally fine-grained network states, and routing decisions. Instead, for this problem, data can be collected by interacting with a network environment, which makes RL a suitable training paradigm if interfacing a suitable network environment with the training framework. Research on non-ML algorithms for TE has previously recognized the need for standardized environments to evaluate algorithms, proposing *REPETITA* (Gay et al., 2017b) and *YATES* (Kumar et al., 2018) to facilitate prototyping and reproducing results. Yet, these frameworks are not designed for interactive data collection for training, and they abstract away from packet-level network dynamics. Alternatively, packet-level network simulators like *ns-3* (Henderson et al., 2008) or *OMNeT++* (Varga & Hornig, 2010) can be combined with inter-process connectors like *ns3-gym* (Gawłowicz & Zubow, 2019) or *ns3-ai* (Yin et al., 2020) to create interactive environments that can used both for training and evaluation. If populated with suitable topologies, traffic data and control decisions, such environments can serve large amounts of high-detail training data. Notable examples include *RL4NET++* (Chen et al., 2023) and *RayNet* (Giacomoni et al., 2024) which build on *OMNeT++*, and *PRISMA* (Alliche et al., 2022), *PackeRL* (Boltres et al., 2024) and *NetworkGym* (Haider et al., 2024) which use *ns-3*. Of these, *RL4NET++*, *PRISMA* and *PackeRL* can train neural routing algorithms while supplying some telemetry features, either on a packet level of detail (*PRISMA*) or aggregated per flow (*RL4NET++* and *PackeRL*). While *RL4NET++* allows for asynchronous action updates per agent, it ignores state propagation and inference delays and uses message passing via *ZeroMQ* (Hintjens, 2013) for simulator-ML communication, which according to Yin et al. (2020) is several times slower than using shared memory. *PRISMA*, too, uses *ZeroMQ* for information exchange, but does respect communication delays by

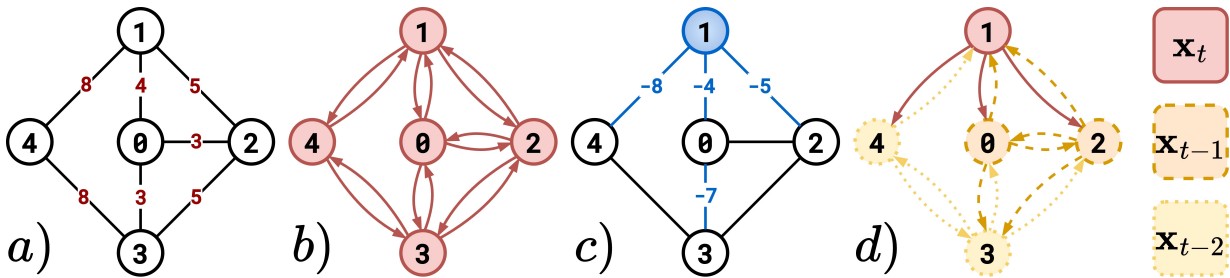

Figure 2: Observation graph assembly for the *mini5* topology, shown in **a)** with link delays in ms. **b)**: The "birds-eye" view of network state $S_t$ contains all current node and edge states $\mathbf{x}_t$. **c)**: We designate node 1 as the central node $v_c$, having the lowest maximum delay of 8 ms to all other nodes. Colored edges show the minimum-delay spanning tree used for communicating state, observation, and action information. **d)**: Observation graph $O_{1,t}$ of node 1 at time $t$. For a step granularity of $\tau = 5$ ms, node states $\mathbf{x}_{0,t}$ and $\mathbf{x}_{2,t}$ will be available to node 1 at time $t + 1$; $\mathbf{x}_{3,t}$ and $\mathbf{x}_{4,t}$ will be available at time $t + 2$.

sending state update packets over the network. Yet, its network model is limited to UDP traffic, dismissing that a large portion of today's internet usage comes from TCP traffic (Schumann et al., 2022). *PackeRL*, on the other hand, is limited in that it provides only global network states, disregarding communication and inference delays. Our own framework is the first to combine the rich network model of *ns-3* with fast inter-process exchange via *ns3-ai*'s shared memory, while providing telemetry-infused network state views that respect communication delays to centralized or decentralized routing agents.

**Summary.** All in all, existing work lacks a delay-aware neural routing algorithm that utilizes telemetry information to respond to traffic bursts with millisecond responsiveness. Existing approaches are telemetry-oblivious (Bhavanasi et al., 2022), too fine-grained to scale (Xiao et al., 2020; Mai et al., 2021), utilize centralized network states that disregard communication delays (Boltres et al., 2024), or use distributed non-coordinating agents (Gui et al., 2024). To train such an algorithm, we also lack a framework with packet-level simulation detail that provides telemetry-enriched network states respecting communication delays. Existing frameworks do not collect telemetry data (Giacomoni et al., 2024; Haider et al., 2024), ignore communication delays (Chen et al., 2023; Boltres et al., 2024), or support only a subset of encountered traffic models (Alliche et al., 2022). To close these gaps, in Section 3 we extend *PackeRL*'s network model with a new implementation for in-band network telemetry and delay-aware state and action communication. Then, in Section 4 we introduce our new neural routing algorithm *LOGGIA*, which fulfills aforementioned algorithm requirements and trains more efficiently than previous approaches.

## 3 Simulation Framework

### 3.1 Network Model

We model our network topologies as undirected graphs, where each router represents a routing node, and pairs of routers are connected by symmetric point-to-point links (i.e. graph edges). In our model, routing nodes act as the traffic control instances of the network, as well as ingress/egress points for network traffic. This abstraction simplifies our simulation while maintaining a full model of the network layers involved in routing, whereas a real-world deployment of our system would separate routers and endpoints (i.e. senders and receivers). Source and destination Internet Protocol (IP) addresses are automatically mapped to the corresponding routers/graph nodes. Each routing node performs destination-based forwarding: For each incoming packet, the router looks up the destination address with the corresponding row in the forwarding table, and forwards the packet to the resulting outgoing port.

We assume that the graph is connected (i.e. mutual reachability between all router pairs) and known to all routing nodes, that the links between nodes transmit data without failures, and that all network operations happen deterministically. We formulate our closed-loop routing problem as an Markov Decision Process (MDP) with discrete time steps and present a formalism in Section 3.3. During operation, each routing node locally collects data via In-Band Network Telemetry (INT) (Tan et al., 2021) for itself, as well as the local

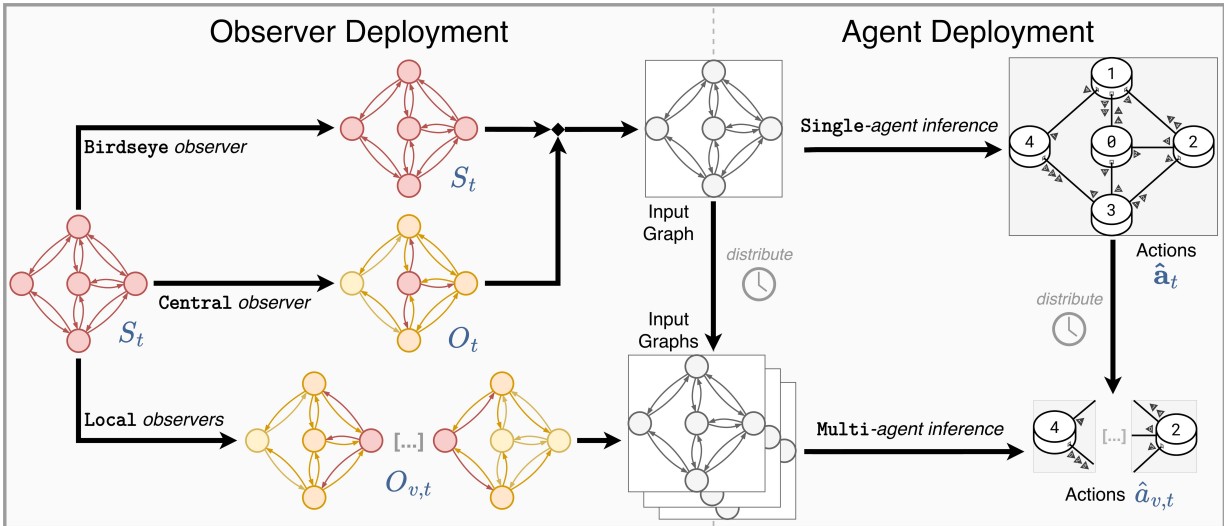

Figure 3: The different possible deployment options within our framework. Each interaction step requires local routing preferences $\hat{a}_{v,t}$ to be installed at the respective routers. **Left:** Observer deployments. The `Birdseye` observer has access to the latest node and edge states $\mathbf{x}_t$ in the network without delay, observing the network state graph $S_t$ as-is. The `Central` observer aggregates $\mathbf{x}_t$ into an observation graph $O_t$ subject to communication delays. `Local` observers each aggregate $\mathbf{x}_t$ from the network, subject to delays, into their own views $O_{v,t}$. **Right:** Agent deployments. The `Single` agent can be a birdseye agent or a central agent, and for the latter, actions $\hat{a}_{v,t}$ must be communicated to routing nodes $v \in V_t$ after inference. In the `Multi`-agent setting, distributed routing agents work with the network state $S_t$ or central observation $O_t$ (the latter incurs delay when $O_t$ is communicated to the local agents), or locally obtained observations $O_{v,t}$.

network interfaces. Combined with topology characteristics like link data rate and delay, at time $t$, this results in node- and edge-level state vectors $\mathbf{x}_{v,t} \in \mathbb{R}^{d_v}$, $\mathbf{x}_{e,t} \in \mathbb{R}^{d_e}$ per node $v \in V_t$ and directed edge $e \in E_t$ in the network graph. The aggregation of such node and edge states, together with the global topology graph and a global state vector $\mathbf{x}_{u,t}$, yields an attributed directed network state graph $S_t = (V_t, E_t, \mathbf{X}_{V_t,t}, \mathbf{X}_{E_t,t}, \mathbf{x}_{u,t})$ at time $t$. As opposed to the undirected network topology graph, $S_t$ is directed because network usage and the resulting telemetry data may be asymmetric.

## 3.2 Communication Model

The state graph $S_t$ depicted in Figure 2b) represents a "birds-eye" view of the network that aggregates localized information from all nodes and edges while ignoring communication delays. Even though previous work assumes such a birds-eye view, it cannot be obtained by any observing component for any real-world network. Figure 2c) and 2d) shows that local information will always arrive with a certain delay at its destination, depending on the delay along the network path. In contrast, our framework provides delay-aware observations of the network state, and the following paragraphs describe the underlying procedure and communication model. In this work, we assume that all state and action communication happens out-of-band and on the shortest-delay paths between all node pairs, which are precomputed for the given network topology. In practice, such meta-data could be sent over dedicated channels to minimize excess delay, and Appendix A.2 discusses the message overhead that would incur in that case.

**Observer and Agent Deployments.** For many networked control problems, it is not trivial to determine when, how and with whom agents should communicate to achieve the best result (Zhu et al., 2024). In our setting, communication depends on whether node and edge states are aggregated locally at each router or at a central control instance, and whether this instance or the individual routing nodes make routing decisions. Both the placement of "observers" and "agents" influence how communication delays have to be factored into the problem formulation. For observers, we conceive three options, as illustrated in Figure 3's left side: The `Birdseye` observer has access to all current node and edge states in the network, i.e, the

network state $S_t$. The `Central` observer aggregates local state information at a single location, subject to communication delays. We select the central node as $v_c = \arg\min_{v \in V} \max_{u \in V} \kappa^*(v, u)$, i.e., the node $v \in V$ with the smallest maximum shortest-path delay $\kappa^*(v, u)$ to any other node $u \in V$ in the network[2]. The result is a central observation $O_t$ of the network state. `Local` observers are placed at each node and aggregate local state information, subject to delays, into their own views $O_{v,t}$. Regarding agent placements, Figure 3's right side shows the two considered agent deployments: Routing actions can be performed by a `Single` agent using the network state $S_t$ or central observation $O_t$, or multiple node-level agents (denoted `Multi`). In the `Multi`-agent setting, distributed routing agents $v \in V_t$ work with the network state $S_t$ if using a `Birdseye` observer, a central observation $O_t$ provided by a `Central` observer, or observations $O_{v,t}$ provided by their own `Local` observers.

Factorizing the three observer and two agent deployment options yields five relevant *deployment modes*: `Birdseye-Single`, `Central-Single`, `Birdseye-Multi`, `Central-Multi`, `Local-Multi`. The combination `Local-Single`, in comparison to `Local-Multi`, includes redundant communications to a central node if using homogeneous agent policies (Section 4.2), and will thus not be considered. The three delay-aware deployment modes `Central-Single`, `Central-Multi`, `Local-Multi` differ in how information exchange incurs delay, and it is not trivial to determine which of these modes work best for a delay-aware neural routing algorithm. Our results in Figures 7 and 18 show that the fully distributed `Local-Multi` deployment mode leads to the best routing performance among all delay-aware deployment modes, which thus becomes the default in our experiments of Section 5. Nevertheless, our experiments cover all the above deployment modes to analyze how they affect routing performance.

**Propagation and Inference Delays.** Figure 2 provides an example for what each observer sees at time $t$, before "freezing time" in our step-based interaction loop to allow for action computation. At this point, we have respected communication delays that occur up to completing the observation process, but not the delays occurring after observation. To fully respect communication and inference delays, for node $u \in V_t$, we delay populating $u$'s forwarding table with new actions by $\kappa_{\text{install}}(u, t)$: For fully decentralized observer-agent deployment (mode `Local-Multi`), $\kappa_{\text{install}}(u, t) = \lambda_{\text{ac}} \kappa_{ac}(u, t)$, where $\lambda_{\text{ac}}$ is a scaling hyperparameter that factors in potential inference speedups attainable by specialized hardware (Zhang et al., 2023; Procaccini et al., 2024), and $\kappa_{ac}(u, t)$ is the inference time taken by routing agent $u$ at time $t$. Additionally, for modes `Central-Single` and `Central-Multi`, we add a second delay term $\kappa^*(v_c, u)$, such that $\kappa_{\text{install}}(u, t) = \lambda_{\text{ac}} \kappa_{ac}(u, t) + \kappa^*(v_c, u)$. In `Central-Multi` mode, $\kappa^*(v_c, u)$ describes the delay that occurs when sending centrally assembled observations $O_t$ to local agents. In `Central-Single` mode, it describes the delay of sending centrally computed routing actions to local nodes for installation.

**Assumptions.** Overall, we model inference delays, and delays that occur when communicating local/aggregated state information and routing decisions. Our interaction loop corresponds to a near-real time system in which time does not stop for state observation and propagation, nor for action computation. Yet, this model is only an approximation for real-world network state communication. It excludes additional potential sources of delay like forwarding table updates, or queuing delays for state information packets. Such delays can also lie in the millisecond range for larger networks (Vissicchio et al., 2015), and future work may add the installation time as a separate factor to the overall action delay $\kappa_{\text{install}}(u, t)$. Moreover, the data overhead for full observation communication messages may exceed the maximum transmittable data size per packet for larger networks, such that a real-world system may not be able to send state updates in every interaction step. We discuss the overhead caused by communicating state and action information across the network in Appendix A.2, and point to the experiment of Appendix D.2 with $\tau = 10$ms for a showcase of neural routing algorithms with reduced control granularity.

### 3.3 Telemetry-Aware Routing as a Sequential Decision Problem

We formulate our closed-loop routing problem as a discrete-time (partially observable) MDP. The base formulation is given as $\mathcal{M}_{\text{Birdseye-Single}} = (\mathcal{S}, \mathcal{A}, \mathcal{Z}, \mathcal{P}_{\mathcal{S}}, \mathcal{P}_{\mathcal{Z}}, r)$ and corresponds to the `Birdseye-Single` deployment mode, with variations for the other modes explained in the following. Our environment is an *input-driven environment* (Mao et al., 2018) with external input variables $\mathbf{z} \in \mathcal{Z}$ describing incoming traffic,

---

[2]If there is a tie between multiple nodes, the node with the lowest node ID is selected.

an input-dependent deterministic transition function $\mathcal{P}_{\mathcal{S}} : \mathcal{S} \times \mathcal{A} \times \mathcal{Z} \to \mathcal{S}$ implemented by the network model, and an input transition kernel $\mathcal{P}_{\mathcal{Z}}(\mathbf{z}_{t+1} \mid \mathbf{z}_t)$ described by our traffic model. In practice, traffic patterns in computer networks can be volatile and unpredictable on a sub-second scale (Wendell & Freedman, 2011), and thus we consider the upcoming traffic $\mathbf{z}_t$ to be unobserved by the agent. The global network state space $\mathcal{S}$ containing directed attributed graphs $S_t$ is only partially observable in delay-aware deployment modes: $\mathcal{M}_{\texttt{Central-Single}}$ and $\mathcal{M}_{\texttt{Central-Multi}}$ include a global observation function $\mathcal{O}$, whereas $\mathcal{M}_{\texttt{Local-Multi}}$ includes per-agent observation functions $\{\mathcal{O}_i\}$. These observation functions factor in the delayed arrival of remote state information at the observer(s). The global action space $\mathcal{A} = \{A \mid A_t = \{(u,z) \mapsto v \mid u, v, z \in V_t, v \in \mathcal{N}_u\}\}$ aggregates a bundle of next-hop neighbor selections $v \in \mathcal{N}_u$ per destination node $z \in V_t$ for all routing nodes $u \in V_t$ in the network. For deployment modes with distributed agents, it can be expressed as an aggregation $\{\mathcal{A}_i\}$ of the local action spaces. Next, the delay-aware formulation $\mathcal{M}_{\texttt{Central-Single}}$, $\mathcal{M}_{\texttt{Central-Multi}}$ and $\mathcal{M}_{\texttt{Local-Multi}}$ contain a modified state transition function $\mathcal{P}_{\mathcal{S}}$ that includes the installation delays $\kappa_{\text{install}}$ described in Section 3.2 above. Finally, the reward function $r : \mathcal{S} \times \mathcal{A} \times \mathcal{Z} \to \mathbb{R}$ rates routing performance based on the chosen network performance metrics. Our goal is to find a policy $\pi : \mathcal{S} \times \mathcal{A} \to [0,1]$ that maximizes the return, i.e., the expected discounted cumulative future reward $J_t := \mathbb{E}_{\pi(\mathbf{a}|\mathbf{s})}\left[\sum_{k=0}^{\infty} \gamma^k r(S_{t+k}, A_{t+k})\right]$. Following Boltres et al. (2024), we optimize for the global delivery rate per step (in MB, also called goodput) and use it as our primary evaluation metric. Optimizing for delivery rate co-optimizes other performance metrics like congestion and latency to an extent, as shown in Tables 3 and 4 of Appendix D.

### 3.4 Framework Implementation

Our framework is implemented in Python and C++, and uses *ns-3* (Henderson et al., 2008) as a simulation backend with packet-level detail and *ns3-ai*'s shared memory (Yin et al., 2020) for fast inter-process communication between simulator and learning loop. We adopt the implementation for state monitoring using INT, interaction logic, and action installation of the *PackeRL* framework (Boltres et al., 2024), and implement new logic to assemble node/edge states to observation graphs as per the deployment modes introduced in Section 3.2. Appendix A.1 provides more implementation details on observation graph assembly. Moreover, we implement logic to accommodate for multi-agent training of our new algorithm *LOGGIA*, which is presented in the following Section 4. The multi-agent extension logic includes observation graph stacks and masking operators that enable batched multi-agent inference and training within our routing formulation, as well as reward functions that provide global and node-level rewards (detailed in Appendix A.3).

## 4 Algorithm

For single-path IP routing, we train a neural network policy $\pi : \mathcal{S} \times \mathcal{A} \to [0,1]$ that turns one or multiple attributed network state graphs into a next-hop neighbor selection per possible pair of routing and destination node. We adopt the two-stage design $\pi_\theta = \psi \circ \zeta_\theta$ used by *MAGNNETO* (Bernárdez et al., 2023) and *M-Slim* (Boltres et al., 2024), which use a GNN parametrized by $\theta$ to output link weights for subsequent shortest-path computations using Dijkstra's algorithm (function $\psi$). Our own approach *LOGGIA* introduces a log-space link-weight parameterization, and combines techniques from core RL research and other application domains to improve performance and reduce variance in more challenging networking environments (c.f. Section 6). Namely, we include an adaptive entropy regularizer inspired by Soft Actor-Critic (SAC)(Haarnoja et al., 2018), an early stopping mechanism based on the Kullback-Leibler (KL) divergence(Kullback & Leibler, 1951; Dossa et al., 2021), and an Imitation Learning (IL) pretraining phase that warm-starts $\pi_\theta$ from expert demonstrations(Ross et al., 2011). To the best of our knowledge, we are the first to apply these techniques in combination to increase training stability of a neural network algorithm in a network routing problem.

### 4.1 Policy Architecture

To support attributed input graphs with edge, node and global features, the first stage $\zeta_\theta$ of our policy uses Message Passing Networks (MPNs) (Gilmer et al., 2017; Sanchez-Gonzalez et al., 2020) . MPNs are a variant of message-passing GNNs that iteratively updates latent node, edge and global features over $L$ steps. According to Bronstein et al. (2021), message-passing GNNs are a superset of convolutional (Kipf & Welling, 2017) or attentional (Veličković et al., 2018) GNNs in terms of representational capacity. With

initial node features $\mathbf{x}_v^0$ (part of node feature matrix $\mathbf{X}_{v,t}$), edge features $\mathbf{x}_e^0$ (part of edge feature matrix $\mathbf{X}_{e,t}$; $e = (v,u)$), and global features $\mathbf{x}_U^0 = \mathbf{x}_{U,t}$, the $l$-th step is given as

$$\mathbf{x}_e^{l+1} = f_E^l(\mathbf{x}_v^l, \mathbf{x}_u^l, \mathbf{x}_e^l, \mathbf{x}_U^l), \qquad \mathbf{x}_v^{l+1} = f_V^l(\mathbf{x}_v^l, \bigoplus_{e'=(v,u'),\ u' \in \mathcal{N}_v} \mathbf{x}_{e'}^{l+1}), \qquad \mathbf{x}_U^{l+1} = f_U^l(\bigoplus_{v' \in V} \mathbf{x}_{v'}^{l+1}, \bigoplus_{e' \in E} \mathbf{x}_{e'}^{l+1}, \mathbf{x}_U^l),$$

(1)

using a concatenation of the features' mean and minimum for the permutation-invariant aggregation $\oplus$. The functions $f_E, f_V, f_U$ are implemented as Multilayer Perceptrons (MLPs), and $f_E, f_V$ use the adjacency matrix $\mathbf{A}$ to compute updates for all edge/node representations in parallel at each step. A readout layer $f_{\mathrm{out}}$ transforms the final edge latent features $\mathbf{x}_e^L$ into two output values $(\mu, \sigma)$ per graph edge. During inference, the link weights are simply obtained as $\exp(\mu)$. This design supports arbitrary computer network topologies out-of-the-box because the MPN operates directly on the attributed graphs obtained from the monitored network. *LOGGIA* improves over existing approaches via two architectural decisions (ablated in Appendix C.1): Firstly, in contrast to *MAGNNETO* (Bernárdez et al., 2023) and *M-Slim* (Boltres et al., 2024), *LOGGIA* works directly on the input graph instead of converting the input graph into its *line digraph* (Harary & Norman, 1960) representation, obtaining the per-link output from the latent node features $\mathbf{x}_e^L$. Secondly, our MPN infers proto-link weights in log-space, whereas *MAGNNETO* uses its neural network output to increment link weights in an iterative manner, and *M-Slim* uses the output space directly in conjunction with a Softplus function.

The second policy stage $\psi$ obtains routing actions $\hat{\mathbf{a}}_t$ from the link weights computed in the first stage by computing shortest paths using Dijkstra's algorithm (Dijkstra, 1959). In single-agent deployment, this is done per source node, analogous to *M-Slim*, with a total runtime complexity of $O(|V|^2 \log |V| + |V||E|)$ (Cormen et al., 2022). In multi-agent deployment, each node $u$ is responsible for designating a next-hop node $v \in \mathcal{N}_u$ per destination node $z \in V_t$, which can be obtained by running single-source Dijkstra in $O(|V| \log |V| + |E|)$ with the locally obtained link weights. Regarding the overall complexity of the inference step, the shortest-paths computation complexity of $\psi$ dominates the MPN's complexity of $O(|V|D)$ (Alkin et al., 2024) in both single- and multi-agent deployment, unless the graph's maximum node degree $D$ is very large ($D \gtrsim \log |V|$). The configuration hyperparameters used by *LOGGIA*'s architecture are detailed in Appendix B.1.

## 4.2 Effective Reinforcement Learning for Telemetry-Aware Neural Routing

We use Proximal Policy Optimization (PPO) (Schulman et al., 2017) to train our policy $\pi_\theta$ and a value function $W_\phi : \mathcal{S} \to \mathbb{R}$, with disjoint parameters $\theta$ and $\phi$, and separate Adam optimizers (Kingma & Ba, 2014). The value function $W_\phi$ comprises an MPN matching our policy's MPN structure, followed by a node-level MLP, a node-level readout layer, and a global pooling operation. PPO's clipped surrogate objective is

$$\mathcal{L}_{\mathrm{clip}}(\theta) = \mathbb{E}_t \left[ \min\left( v_t(\theta), \hat{A}_t,\ \mathrm{clip}\left(v_t(\theta), 1 - \epsilon, 1 + \epsilon\right), \hat{A}_t \right) \right], \quad v_t(\theta) = \frac{\pi_\theta(a_t \mid s_t)}{\pi_{\theta_{\mathrm{old}}}(a_t \mid s_t)}.$$

(2)

**Efficient Policy Learning.** We adjust PPO's policy training in two aspects. Firstly, we incorporate entropy-regularized exploration inspired by SAC (Haarnoja et al., 2018), augmenting the policy objective with an entropy bonus and learning an adaptive temperature: We minimize $\mathcal{L}_{\mathrm{clip}}(\theta) + \alpha \mathbb{E}_t[\mathcal{H}(\pi_\theta(\cdot \mid s_t))]$, and update $\alpha$ via a dedicated optimizer using $\alpha = \mathrm{softplus}(\rho)$ with trainable $\rho$ and temperature loss

$$\mathcal{L}_\alpha(\rho) = \mathbb{E}_t \left[ \alpha, \left( \mathcal{H}_{\mathrm{targ}} - \mathcal{H}(\pi_\theta(\cdot \mid s_t)) \right) \right],$$

(3)

where $\mathcal{H}_{\mathrm{targ}}$ is a target entropy hyperparameter. Secondly, we apply early stopping for policy updates. After each epoch, we estimate the policy clip fraction as $\hat{p}_{\mathrm{clip}} = \mathbb{E}_t[\mathbf{1}\{v_t(\theta) \notin [1 - \epsilon, 1 + \epsilon]\}]$, with $\mathbf{1}$ denoting the indicator function, and the mean KL divergence (Kullback & Leibler, 1951) as $\widehat{\mathrm{KL}} = \mathbb{E}_t[\mathrm{KL}(\pi_{\theta_{\mathrm{old}}}(\cdot \mid s_t) || \pi_\theta(\cdot \mid s_t))]$. If $\hat{p}_{\mathrm{clip}} > \delta_{\mathrm{clip}}$ or $\widehat{\mathrm{KL}} > \delta_{\mathrm{KL}}$ or (with $\delta_{\mathrm{clip}}, \delta_{\mathrm{KL}}$ being hyperparameters) after an update epoch, we deactivate further policy optimization for that training step, while continuing value-function optimization with $\phi$ as usual. Our early stopping mechanism is a lightweight alternative to

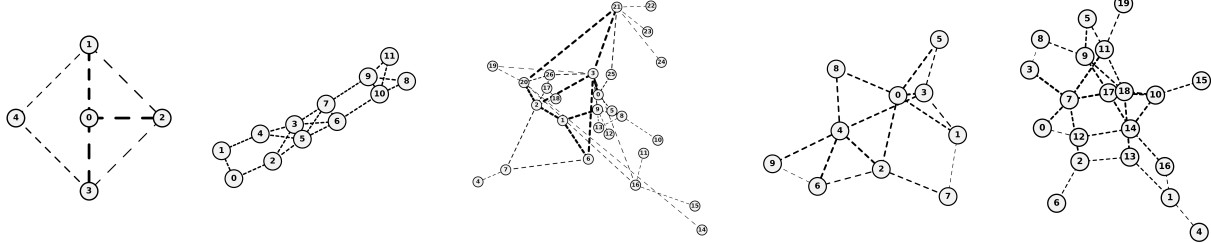

Figure 4: Network topologies used in our experiments. From left to right: *mini5*, *B4* (version used in Xu et al. (2023)), *GEANT* (2001 version obtained from the Topology Zoo, Knight et al. (2011)), *nx-XS* (example), *nx-S* (example). Thicker edges denote larger link datarate, fewer dashes denote lower link latency.

information-theoretic trust-region mechanisms based on the KL divergence (Kakade, 2001; Schulman et al., 2015). Despite its simplicity and the absence of trust-region guarantees, this mechanism increases training stability in practice (Dossa et al., 2021), as shown in Ablation C.2. The configuration hyperparameters used by our PPO implementation are detailed in Appendix B.2.

**Multi-Agent Training.** The multi-agent deployment modes `Birdseye-Multi`, `Central-Multi`, and `Local-Multi` suggest the use of multi-agent PPO algorithms for training. We extend our PPO implementation to support homogeneous multi-agent training with shared policy parameters and a centralized value function. Our implementation aligns with Multi-Agent PPO (MAPPO) (Yu et al., 2022) and uses per-agent rewards obtained from local event tracing (implementation details are provided in Appendix A.3). It can be configured to use observations provided from a single `Birdseye`/`Central` observer, or `Local` observers, realizing aforementioned multi-agent deployment modes. Our results in Figure 16 of Appendix C.5 show that, interestingly, a `Central` training observer achieves the best results when paired with per-agent rewards, despite evaluation in the `Local-Multi` deployment mode. Therefore, by default, all our training algorithms use `Central` observers by default.

### 4.3 Improving Training Stability with Imitation Learning Pretraining

As deep RL notoriously suffers from training instabilities due to the bias-variance tradeoff and the need for exploration in training, researchers have leveraged Imitation Learning (IL) for routing in the TE setting (Guo et al., 2025), as well as other research domains (Silver et al., 2016; Aravind et al., 2018; Foster et al., 2024). When expert demonstrations are available, IL algorithms can train deep neural networks to imitate these demonstrations. For this, however, the expert demonstrations have to fit the formalism of the underlying decision problem. Aside neural routing baselines, we can leverage static routing algorithms, such as shortest-path computations with topology-dependent path costs, and warm-start our neural routing algorithm by training it to imitate the static baseline regardless of the current telemetry information. Our IL algorithm collects training data by interacting with our training environment in the same manner as our PPO algorithm, plus additional expert actions for every interaction step. The latter resembles the human expert data collected and used in *DAgger* (Ross et al., 2011). We minimize a mean-squared error $\mathcal{L}_{\mathrm{IL}} = \frac{1}{m} \sum_{i=1}^{m} \left( \tilde{\mathbf{w}}_i^{(s)} - \tilde{\mathbf{w}}_i^{(e)} \right)^2$ between the normalized student link weights $\tilde{\mathbf{w}}^{(s)}$ and normalized expert link weights $\tilde{\mathbf{w}}^{(e)}$, where normalization means dividing each set of link weights by its own mean to make the objective scale-invariant. Additional configuration parameters are described in Appendix B.2. While not sufficient for standalone training, our results in Ablation C.4 show that IL can serve as a pretraining phase that improves subsequent PPO training phases. An alternative IL approach that forgoes environment interaction, however, yields inferior results.

## 5 Experiments

Our experiments seek to answer the following questions: **(i)** Do neural routing algorithms outperform non-neural alternatives when evaluated in a near real-time setting? **(ii)** Do communication and inference delays

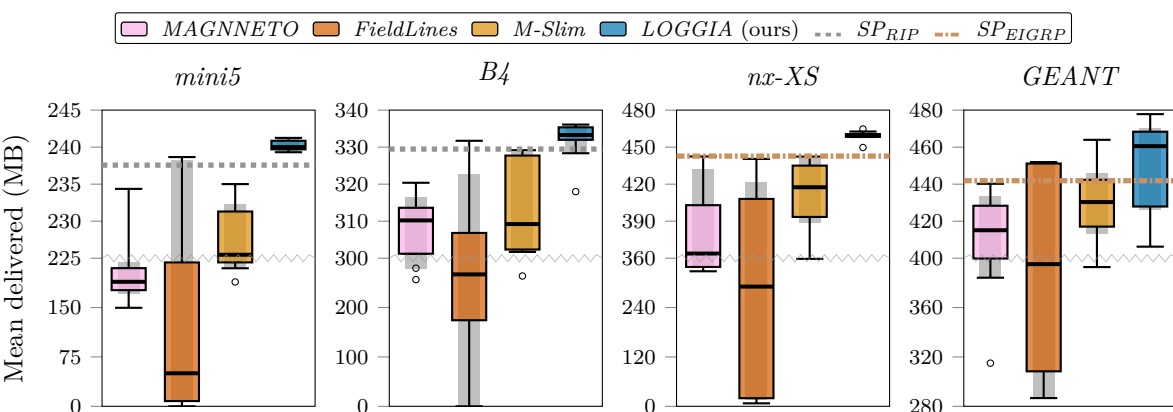

Figure 5: Performance of baseline algorithms and our algorithm *LOGGIA* for various topology presets, evaluated in delay-aware deployment. Dashed lines denote the best *SP* baseline per evaluation. Semi-transparent bars overlaid on the boxplots represent the 95% bootstrapped confidence intervals. Above the zigzag line, the y-axis is enlarged for better readability. In the more realistic near-real time routing setting, all neural routing algorithms except for our approach *LOGGIA* fail to outperform static shortest-path routing. Moreover, *LOGGIA* shows lower performance variance for most topology presets.

impact convergence during training? **(iii)** In the near real-time setting, should network observers and routing agents be deployed centrally, or distributed to each router?

**Network Topologies.** We train and evaluate on different synthetic and real network topologies illustrated in Figure 4. *mini5* is a small synthetic five-node network. The *B4* topology (Jain et al., 2013) represents Google's Inter-Datacenter Network (12 nodes, 17 links), whereas the *GEANT* topology models a network interconnecting research and education institutions in Europe (we use the 2001 version obtained from the Topology Zoo, Knight et al. (2011), with 27 nodes and 38 links). The *nx* topology family defined in Boltres et al. (2024) provides synthetic topologies of varying size classes, e.g. *nx-XS* with 6 to 10 nodes and *nx-S* with 11 to 25 nodes. Link capacities and latencies are scaled to lie between 50 and 200 Mbps and 1 and 10 ms for all topologies, respectively. Packet buffer sizes are calculated as the product of the incident link's capacity and round-trip time. Figures 19 and 20 provide additional visualizations for the topologies.

**Networking Episodes.** We simulate two seconds of continuous network operation per episode, each split into 400 time steps of $\tau = 5$ ms each. During each episode, we keep the network topology unchanged and inject synthetic traffic flows into the simulator backend, which are generated akin to Boltres et al. (2024). These include 80% TCP flows and 20% constant-bitrate UDP flows, with arrival times and flow sizes drawn from distributions that resemble measurements in real-world data center networks (Benson et al., 2010; Vargas et al., 2019). We scale the generated flow inter-arrival times, and thus the intensity of the injected traffic, such that packet drops are unavoidable for static routing.

**Baselines.** We compare against routing algorithms computing shortest paths with predefined link cost metrics, and dynamic algorithms using neural networks. A prerequisite for comparison is that the baseline implements a function $f : \mathcal{S} \to \mathcal{A}$ that matches our MDP formulation. While OSPF (Moy, 1997), EIGRP (Savage et al., 2016), and RIP (Hedrick, 1988) implement routing logic beyond $f$, their path calculation can be approximated by a shortest-path algorithm with protocol-specific link weight metrics. We thus denote the baseline using OSPF's datarate-dependent link metric as $SP_{OSPF}$, the baseline using EIGRP's default datarate-/delay-dependent link metric as $SP_{EIGRP}$, and the baseline that implements lowest-hop-count routing akin to RIP as $SP_{RIP}$[3]. Since network topologies remain unchanged within an evaluation episode, the paths provided by these baselines are static. For dynamic neural baseline algorithms, we consider the algorithms *FieldLines* and *M-Slim* from Boltres et al. (2024), and *MAGNNETO* from Bernárdez et al. (2023).

---

[3]The performance plots of Section 6 and Appendix C and D show only the best performing *SP* baseline per evaluation.

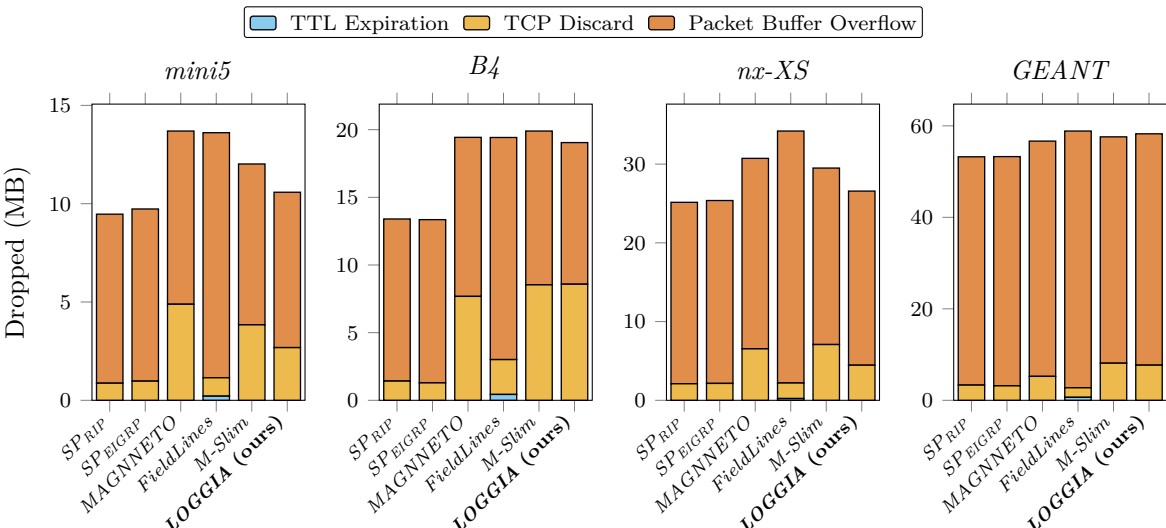

Figure 6: Data dropped by cause per algorithm (*LOGGIA*, shortest-path baselines, neural baselines) for various topology presets, evaluated in delay-aware deployment. For *LOGGIA* and the neural baselines, columns show the mean values across 8 random seeds. Overall, static shortest-paths routing lead to the lowest data drops, with *LOGGIA* being the best of the neural routing algorithms. It achieves comparable overall drop numbers in all topology settings except for *B4*, and generally drops slightly less data due to overflowing packet buffers than other approaches. Except for *FieldLines*, all neural routing algorithms notably increase the amount of data dropped due to TCP discards, which includes out-of-order packet arrivals. *FieldLines*, on the other hand, is the only algorithm that drops packets due to Time-to-Live (TTL) expiration, as non-coalescent paths may form routing loops that lead to packets reaching the maximum hop count (set to 64).

**Training and Evaluation Details.** We train every neural routing algorithm on 8 different random seeds, and the neural baselines according to their respective training setups. We train our own algorithm *LOGGIA* for 10 training IL iterations, followed by 10 PPO/MAPPO iterations. Each training iteration consists of 16 episodes, which either includes four different topologies with four different traffic sequences each (for the *nx* topology family), or 16 different traffic sequences in the case of a constant network topology (*mini5*, *B4*, *GEANT*). In total, this yields 128000 training steps and is sufficient to reach convergence. For IL, we use $SP_{EIGRP}$ as the expert algorithm. Training run times depend on the network topology, traffic characteristics, temporal decision granularity, trainer combination, and chosen hardware. As the number of nodes quadratically influences the memory required for multi-agent PPO training (state graph size and number of agents), we do not use multi-agent training for *GEANT*. Training takes 4 to 36 hours on 2 cores of an Intel Xeon 6780E CPU, largely depending on the scale of network topologies and the intensity of injected traffic. We set the inference delay scaling parameter $\lambda_{ac}$ to 0.2 for all delay-aware observer/agent deployment modes. Unless noted otherwise, we report performance results aggregated over 30 evaluation episodes in the delay-aware setting with fully distributed observers and agents (deployment mode `Local-Multi`), except for *MAGNNETO* which is evaluated in `Central-Single` deployment as its algorithm requires globally aggregated traffic matrices. Each evaluation episode includes a previously unseen traffic sequence, and, for evaluations on the *nx* topology family, a previously unseen network topology. We report the episodic goodput/delivered data in MB, which is our optimization criterion, and supply results on other metrics for the first experiment in Appendix D.1.

## 6 Results

This section presents the main findings of our experiments on near-real time routing control, demonstrating the capabilities of our algorithm *LOGGIA* presented in Section 4. We provide supplementary parameter studies in Appendix C, covering *LOGGIA*'s architecture in Appendix C.1, alternative policy and value

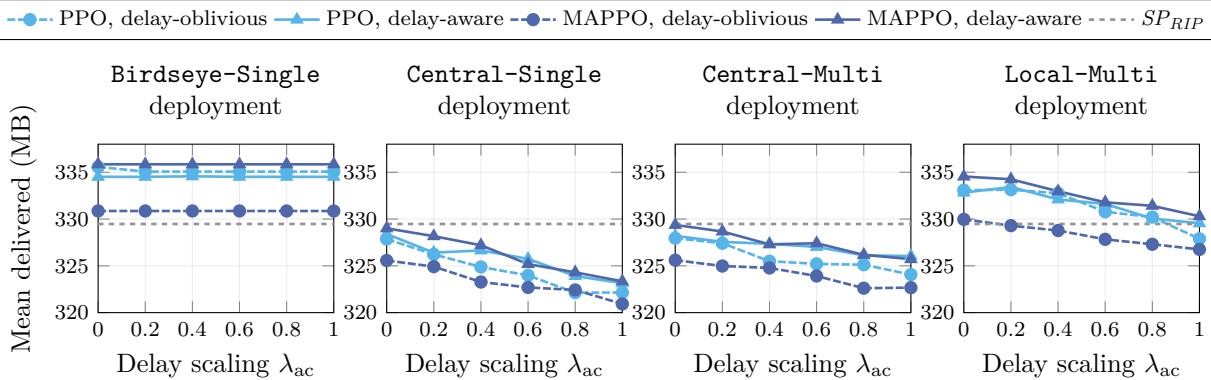

Figure 7: Performance of *LOGGIA* trained with PPO/MAPPO in delay-aware/delay-oblivious configuration, and evaluated in different deployment modes on the *B4* topology. Solid lines show the inter-quartile mean across 8 random seeds per approach. Dashed lines denote the best *SP* baseline per evaluation. All runs include IL pretraining and are evaluated for varying inference delay scalings $\lambda_{ac} \in [0, 1]$ (x-axis per plot). Including communication and inference delays in the interaction model decreases overall performance, as seen in the comparison between the delay-oblivious `Birdseye-Single` and the other delay-aware deployment modes. Of the delay-aware deployment modes, the `Local-Multi` mode with local observers and agents yields the best results, and it is the only deployment mode in which *LOGGIA* outperforms the baseline. Moreover, routing performance decreases with an increasing delay scaling factor $\lambda_{ac}$, i.e., higher inference delays. Finally, respecting delays during training yields superior routing algorithms for multi-agent training, while showing no clear effect for single-agent training.

learning mechanisms in Appendix C.2, path-level PPO exploration mechanisms in Appendix C.3, the choice of training algorithm in Appendix C.4, and alternative multi-agent PPO settings in Appendix C.5. Moreover, Appendix D provides additional results that accompany the following findings.

## 6.1 Routing Performance in the Near-Real Time Setting

Figure 5 shows the routing performance of each neural routing algorithm on the topology presets introduced in Section 5, plus the *SP* baseline that respectively yielded the best results per evaluation. We find that, except for *LOGGIA*, all neural routing algorithms deliver significantly less data per episode than the respective *SP* baseline. In contrast, our algorithm *LOGGIA* visibly outperforms other routing algorithms and converges more stably than the neural baselines, shown by the smaller boxplot whisker spread and confidence interval size for the *mini5*, *B4*, and *nx-XS* topology presets. Meanwhile, *LOGGIA* maintains comparable levels of data loss, as shown in Figure 6. While static shortest-paths routing lead to the lowest drop numbers, *LOGGIA* maintains the lowest overall drop numbers of all neural routing algorithms. It achieves competitive numbers in all topology settings except for *B4*, and generally drops slightly less data due to overflowing packet buffers than other approaches, including the shortest-path baselines. Yet, we also observe that, except for *FieldLines*, all neural routing algorithms notably increase the amount of data dropped due to TCP discards, which includes out-of-order packet arrivals. Supplementary results for this experiment can be found in Appendix D.1 and Appendix D.2: Appendix D.1 provides results for other routing performance metrics like latency, and shows that *LOGGIA*'s performance with respect to other performance metrics is on-par with its competitors. Appendix D.2 shows that the performance of neural routing algorithms depends on environment settings like the permitted control granularity and packet buffer sizes.

## 6.2 The Influence of Communication and Inference Delays

We train *LOGGIA* in single-agent or multi-agent mode, and either respect or ignore communication and inference delays during training. During evaluation, we vary the inference delay scaling parameter $\lambda_{ac}$ in $[0, 1]$, and the four observer-agent deployments `Birdseye-Single`, `Central-Single`, `Central-Multi`, `Local-Multi` introduced in Section 3.2, to investigate their influence on routing quality. The results in

Figures 7, and 18 of Appendix 6.2, show the following: **(i)** There is a notable performance difference between the diagnostic centralized `Birdseye-Single` deployment mode, which ignores all sources of delay, and the deployment modes `Central-Single`, `Central-Multi`, `Local-Multi`, which respect communication delays. **(ii)** Of these deployments, the `Local-Multi` mode, in which both observers and agents are deployed locally, yields the best results, and it is the only delay-aware deployment mode in which our algorithm outperforms the baseline. **(iii)** In addition to the communication delays, the delay caused by action inference also influences routing performance in evaluation, which is shown by the decreasing performance for the delay-aware deployment modes `Central-Single`, `Central-Multi`, `Local-Multi` as $\lambda_{ac}$ increases. While *LOGGIA* is faster than baseline algorithms, additional results presented in Appendix D.4 show that the inference time, and thus the routing performance, also depends on the underlying hardware, which exposes a difficulty in evaluating the quality of our policy design and training algorithm individually. **(iv)** Respecting delays during training results in better routing performance for multi-agent training, but shows no clear effect for single-agent training.

## 6.3 Generalization and Scaling

To assess the generalization and scaling capabilities of our algorithm *LOGGIA*, we train on different single topologies or topology groups, and evaluate on the *nx* topology family with node counts of 6 to 10 nodes (*nx-XS*), 11 to 25 nodes (*nx-S*), 26 to 50 nodes (*nx-M*), and 51 to 100 nodes (*nx-L*). Figure 8 shows the evaluation results for the delay-agnostic `Birdseye-Single` deployment mode and the delay-aware `Cnetral-Single` and `Local-Multi` deployment modes, as well as *LOGGIA*'s inference times for `Single`- and `Multi`-agent deployment. The performance results are reported relative to $SP_{RIP}$, the best performing baseline in all evaluated settings of this experiment. We observe that *LOGGIA* is capable of scaling to large topologies as well as the shortest-path baseline, both for the delay-agnostic `Birdseye-Single` deployment mode and the delay-aware `Local-Multi` mode. Yet, the evaluation performance depends on the topology (family) chosen for training: choosing the *B4* topology for training results in inferior performance on all but the smallest evaluation topologies. Interestingly, the results for *mini5* show that it is enough to train on just one topology, showing even slightly superior scaling properties for the largest evaluated topology set *nx-L*. Finally, we note that inference times increase faster for `Single` agent deployment, which is due to the required centralized all-pairs shortest paths computation. The exploding inference times significantly impact *LOGGIA*'s scaling capabilities in the `Central-Single` deployment mode.

## 6.4 Discussion

All in all, we find in Section 6.1 that, whereas baseline algorithms struggle, *LOGGIA* can be trained to deliver more data than well established routing algorithms in our more realistic delay-aware networking setup (Figure 5). It benefits from fine-grained routing control cycles and adequately sized packet buffers (Figure 17), and reduces run-to-run variance in most evaluated setups. From Section 6.2, we infer that our more realistic delay-aware networking environment degrades the routing performance of neural routing algorithms. Together with the finding that *LOGGIA* is the only neural routing algorithm that outperforms shortest-path baselines, this implies that neural routing algorithms should be evaluated in (one of) the delay-aware deployment modes `Central-Single`, `Central-Multi`, `Local-Multi`, to properly assess their routing performance. Section 6.2 further shows that smaller inference times, and the right deployment mode, minimize the negative impact of communication and inference delays. Concretely, the fully distributed `Local-Multi` deployment mode, in which each routing node observes the network and infers routing actions at the same location, provides better routing performance than the `Central-Single` or `Central-Multi` deployment modes which use a central observer node that adds another communication stage to the overall control loop. Crucially, as shown in Figures 7 and 8, it is often the only deployment mode in which our algorithm *LOGGIA* outperforms static routing algorithms based on shortest paths. On the other hand, the effects of communication and inference delays on *LOGGIA*'s training convergence depend on whether using multi-agent or single-agent PPO. For single-agent PPO, including delays during training shows no clear improvement in delay-aware evaluation, as opposed to multi-agent PPO where delay-aware training visibly improves downstream performance. Interestingly, Figure 16 shows that combining multi-agent PPO with a `Central` observer delivers equal if not slightly superior performance, suggesting that local observers/the `Local-Multi` deployment

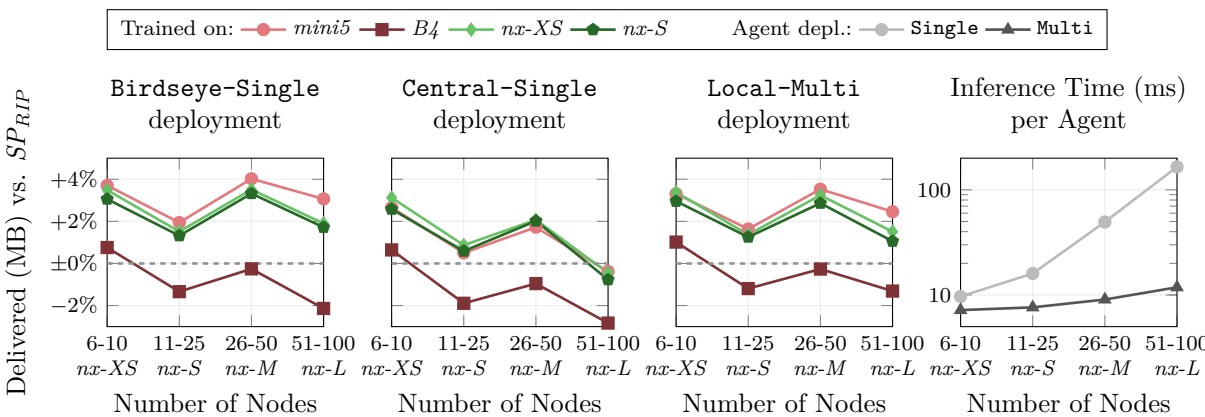

Figure 8: Performance and generalization capabilities of *LOGGIA* trained with delay-aware MAPPO, relative to the best baseline $SP_{RIP}$, evaluated on the *nx* family of topologies with increasing node counts. Solid lines show the inter-quartile mean across 8 random seeds per approach. The choice of training topology set matters, as training on *B4* results in notably worse performance. Varied topologies are not needed, as training on the five-node *mini5* topology already leads to competitive performance. The rightmost plot shows *LOGGIA*'s inference times per agent in `Single` and `Multi` agent deployment mode. The all-pairs shortest path computation required in `Single` agent deployment leads to rapidly increasing inference times, out-scaling GNN inference times. The effect becomes visible in the delay-aware `Central-Single` deployment mode, where routing performance declines with increasing topology sizes.

mode is not required for training. In any case, the performance differences between single- and multi-agent PPO training are small in most evaluated scenarios. The limited benefit of multi-agent training in our routing environment is in line with findings in other cooperative multi-agent settings (Yu et al., 2024; Geng et al., 2025). Moreover, given the quadratically scaling storage and compute requirements for multi-agent training in our framework, single-agent training remains the only alternative for larger training topologies like *GEANT*. From Section 6.3, we infer that *LOGGIA*'s design scales to arbitrary network topologies and topology sizes if choosing the right training topology, matching the scaling properties of shortest-path baselines. Its shortest-path computations, however, slow down inference with increasing network node counts. This becomes particularly evident in the `Central-Single` deployment performance depicted in Figure 8, which requires a centralized all-pairs shortest-path computation. `Local` agent deployment improves the mean inference time by distributing shortest-paths computations, and the increase of observation delays with increasing network sizes appears to be negligible, as shown by the performance similarity between the delay-agnostic `Birdseye-Single` and the delay-aware `Local-Multi` deployment in Figure 8. In any case, it appears that the performance of neural routing algorithms in our delay-aware networking setting crucially depends on short inference times. The inference times, in turn, depend on access to powerful hardware, minimized interference with concurrent computations, and efficient model deployment. Improving inference times of ML components deployed in real-time systems is an ongoing stream of research (Zheng et al., 2015; Procaccini et al., 2024), and our own experimental results of Appendix D.4 suggest that faster hardware, in our case CPUs, indeed increases routing performance.

## 7 Conclusion

In this work, we take a step towards telemetry-aware neural routing algorithms that react to traffic bursts in near-real time. We introduce a delay-aware packet-level simulation framework and *LOGGIA*, a neural routing algorithm that combines graph-based link-weight prediction in log-space, Imitation Learning pretraining, and on-policy Reinforcement Learning. Our results show that, once communication and inference delays are treated as part of the routing problem, as required for realistic routing scenarios, neural baselines no longer outperform static shortest-path routing. In contrast, *LOGGIA* remains effective when using fully distributed network observers and routing agents, delivering more traffic than established routing protocols. Our experiments further show that *LOGGIA*, when trained on just a five-node topology, is able to generalize

to unseen network topologies of up to 100 nodes, consistently outperforming shortest-path baseline algorithms and matching their scaling capabilities. Interestingly, centralized observers lead to inferior performance during evaluation compared to fully distributed deployment, but remain a viable alternative for the training phase. All in all, we find that our delay-aware communication model and *LOGGIA* are an important step towards deploying neural routing algorithms in real-world conditions, as they align with the timing, observability, and actuation constraints of the underlying closed-loop control problem.

**Limitations.** Future work may extend *LOGGIA*'s capabilities to devise multiple paths for routing (Singh et al., 2015), or to support more complex decision policies similar to, for example, the one used by the Border Gateway Protocol (Lougheed & Rekhter, 1990), instead of computing least-cost paths from a single metric. Furthermore, while our communication model supports state/action propagation delays and inference delays, it assumes an out-of-band propagation with unlimited bandwidth, and disregards other sources of delay like the (re-)population of forwarding tables or in-band telemetry operations. Lastly, some of the introduced Markov Decision Process (MDP) formulations introduce delay-awareness into our telemetry-aware closed-loop routing problem by modifying a discrete-time delay-oblivious MDP. Future work may link our formalism to networked (Adlakha et al., 2012), delay-aware (Katsikopoulos & Engelbrecht, 2003) and real-time (Ramstedt & Pal, 2019) MDP formulations, and derive improved theory and algorithms for our setting.

### Broader Impact

The setting considered in this paper is an example for near-time closed control loops that leverage a state monitoring system yielding attributed state graphs and a neural algorithm to transport quantities through a network. Besides Internet Protocol (IP) network routing, we recognize such patterns in road and railway networks, power grids, supply-chain systems, and other critical infrastructures in which decisions must continuously adapt to an evolving network state. We posit that our work can serve as inspiration in all such settings. Nevertheless, neural decision systems introduce additional operational risks: their internal logic is often opaque, rendering validation and accountability difficult. Moreover they may be vulnerable to exploitation or misuse, e.g. through manipulated inputs or deployment outside their intended scope. Neural routing algorithms must therefore remain subject to the hard operational constraints of the respective routing domain, with appropriate safeguards and perhaps human oversight.

### Reproducibility Statement

The code for our routing algorithm *LOGGIA*, the training and evaluation framework, and scripts to reproduce and visualize our results, will be released upon acceptance.

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

## A  Framework Details

### A.1  Monitoring Snapshots and Observation Graph Assembly

Our framework adopts the monitoring capabilities of *PackeRL* (Boltres et al., 2024), which utilizes *ns-3*'s `FlowMonitor` module (Carneiro et al., 2009) and custom event tracing logic to provide a global monitoring snapshot after every simulation step. For any traced event, we know the location the event has been triggered at, the current state of the affected node/link, and the contents of the affected packet's header. This information allows us to create localized `Node`/`EdgeSnapshot` objects, implemented as in Figure 9. For monitored nodes, the event information is used to populate maps in each `NodeSnapshot` which contain usage statistics per source/destination node (Figure 9 right side). The contents of these maps can be used to create traffic matrices that serve as input for baseline routing algorithms. For monitored edges, we split the snapshot information of an edge into an *outgoing* part and an *imcoming* part (Figure 9 left side). Logically, the `OutgoingEdgeSnapshot` contains information regarding the "start" of the underlying link, including the outbound packet buffer and drops happening between buffer dequeueing and transmission. It also includes the link's base information like data rate and delay. On the other hand, the `IncomingEdgeSnapshot` contains information regarding the "end" of the edge, including the receive buffer. We assume edge snapshot information to be instantly visible to the corresponding incident nodes, i.e., the contents of an `OutgoingEdgeSnapshot` that describes an underlying link in direction $u \to v$ is instantly visible to $u$, and the contents of an `IncomingEdgeSnapshot` that describes an underlying link in direction $u \to v$ is instantly visible to $v$. After each simulation step, our framework stores snapshots for every node and edge in its own map, together with the respective monitoring timestamp.

```
typedef struct OutgoingEdgeSnapshot                              typedef struct NodeSnapshot
{                                                                {
    uint64_t channelDataRate;                                        uint32_t nodeId;
    uint32_t channelDelay;
                                                                     unordered_map<uint32_t, uint64_t> txPerReceiver;
    uint32_t txQueueLoad;                                            unordered_map<uint32_t, uint64_t> reTxPerReceiver;
    uint32_t txQueueCapacity;                                        unordered_map<uint32_t, uint64_t> rxPerSender;
                                                                     unordered_map<uint32_t, uint64_t> deliveredPerSender;
    uint64_t transmitted;                                            unordered_map<pair<uint32_t, uint32_t>, double, pair_hash> ttlDroppedPerPair;
    uint64_t transmittable;                                          unordered_map<uint32_t, uint64_t> tcpDiscardedPerSender;
    uint64_t macTxDropped;
                                                                     unordered_map<uint32_t, uint64_t> cumulTxPerReceiver;
    bool isLinkUp;                                                   unordered_map<uint32_t, uint64_t> cumulDeliveredPerSender;
                                                                     unordered_map<pair<uint32_t, uint32_t>, double, pair_hash> cumulTtlDroppedPerPair;
} OutgoingEdgeSnapshot;                                              unordered_map<uint32_t, uint64_t> cumulTcpDiscardedPerSender;

typedef unordered_map<uint32_t, OutgoingEdgeSnapshot> OutgoingEdgeSnapshotMap;
                                                                     unordered_map<uint32_t, double> maxDelayPerSender;
                                                                     unordered_map<uint32_t, double> delaySumPerSender;
                                                                     unordered_map<uint32_t, uint32_t> delayCountPerSender;
                                                                     unordered_map<uint32_t, double> jitterSumPerSender;
                                                                     unordered_map<uint32_t, uint32_t> jitterCountPerSender;

typedef struct IncomingEdgeSnapshot                                  unordered_map<uint32_t, double> tcpSenderSrttSumPerReceiver;
{                                                                    unordered_map<uint32_t, uint32_t> tcpSenderSrttCountPerReceiver;
    uint64_t phyRxDropped;
    bool isLinkUp;                                                   unordered_map<uint32_t, uint32_t> udpRemainingFlowsPerReceiver;
} IncomingEdgeSnapshot;                                              unordered_map<uint32_t, uint64_t> udpRemainingBytesPerReceiver;
                                                                     unordered_map<uint32_t, uint64_t> udpDatarateSumPerReceiver;
typedef unordered_map<uint32_t, IncomingEdgeSnapshot> IncomingEdgeSnapshotMap;   unordered_map<uint32_t, uint32_t> tcpRemainingFlowsPerReceiver;
                                                                     unordered_map<uint32_t, uint64_t> tcpRemainingBytesPerReceiver;
                                                                     unordered_map<uint32_t, uint64_t> tcpCWndSumPerReceiver;

                                                                     OutgoingEdgeSnapshotMap outgoingEdgeSnapshots;
                                                                     IncomingEdgeSnapshotMap incomingEdgeSnapshots;

                                                                 } NodeSnapshot;
```

Figure 9: Our implementation of node and edge snapshots.

To implement our delay-aware observation graph assembly, our framework centrally stores all node snapshots in a map `snap`, indexed by the timestamps $\tau(t)$ of the respective snapshot creation. For a discrete environment step $t$, $\tau(t)$ marks the continuous simulation timestamp of that step. To assemble observations $O_{v,t}$, observing nodes $v$ query the stored snapshots for each node $u \in V_t$ at $\tau(t) - \kappa^*(v, u)$, where $\kappa^*(v, u)$ denotes the communication delay from node $u$ to node $v$ as per their shortest-delay path. Observers $v$ receive the most

recent snapshot stored in the snapshot map, i.e., $\arg\max_{\tau < \tau(t) - \kappa^*(v,u)} \mathtt{snap}(u, \tau)$. From the collected node snapshots, each observer $v$ creates one feature vector per node, per edge, and globally, that make up the network observation $O_{v,t}$. Figure 10 shows the implementation for the resulting feature vectors, which each contain an additional field `observerId` to facilitate batched training and inference on the Python side of our framework. In addition to in-network observers, at time $t$, we obtain a "birds-eye" view of the network state $S_t$ by aggregating all node snapshots available at $\tau(t)$. The corresponding "global" observer is marked by `observerId` $= -1$.

## A.2  State and Action Communication Overhead

The overhead for communicating state information throughout the network can be quantified by determining the payload size of the transmitted node snapshots. With the components shown in Figure 9, and the necessity to store key and value per entry in the unordered maps, we get `payload_size(NodeSnapshot`$_v$`)` $= 32|V|^2 + 62|\mathcal{N}_v| + 232N + 4$ bytes, assuming that `payload_size(uint32_t) = 4` bytes, `payload_size(uint64_t) = 8` bytes, `payload_size(double) = 8` bytes, and `payload_size(bool) = 1` bytes. This, however, includes the data required to assemble detailed traffic matrices used by baseline routing algorithms. A more compact version of the `NodeSnapshot` containing the aggregates of the respective maps would reduce the payload to $62|\mathcal{N}_v| + 268$ bytes. Assuming a data packet can carry up to 1500 payload bytes, a `NodeSnapshot`$_v$ of node $v$ thus fits into a single packet if $v$ has $\mathcal{N}_v < 20$ neighbors. As mentioned in Section 3.2 and depicted in Figure 2, our communication model assumes out-of-band data transmission along the minimum-delay spanning tree to the observing node(s). Locally observed information is sent at the end of each time step. Consequently, for a central observer, $|V| \cdot$ `payload_size(NodeSnapshot)` bytes of information are sent to the observing location. For local observers, this happens $|V| - 1$ times, since every node needs to send its information to all other nodes in the network. We find that, with this communication model, the overhead of state communication can dominate larger network topologies. Therefore, extending our model to in-band communication may require the integration of message compression techniques as proposed in Mostaani et al. (2022) or Yu et al. (2024). Transmitting routing actions across the network is required only for the case of centralized agent deployment. In that case, each routing node receives a vector of selected next-hop neighbors per possible destination node $z \in V$, resulting in a communication overhead that is linear in $|V|$.

## A.3  Global and Local Rewards, Reward Mixing

We implement a reward module as part of our monitoring engine that yields $|V|+1$ reward values after every simulation step – one per routing node, plus a global reward value computed as the sum of MB received by all nodes in the past time step. We obtain node-level rewards per step by tracing "terminal" packet events, i.e., successful packet deliveries, drops, or TCP discards (illustrated in Figure 11). Each terminal packet event provides a base value as reward/penalty that corresponds to the packet's payload size in MB. This value is added to a running reward counter per node on the path taken by the packet, using a decay factor of $\lambda_{\text{decay}} = 0.8$ per path hop for up to three hops as a spatial reward diffusion mechanism. We blend local and global rewards during rollout to facilitate using both reward sources for single-agent and multi-agent Proximal Policy Optimization (PPO). For single-agent PPO, the reward scalar is obtained as $(1-\lambda_R)\bar{r}_{\text{local}} + \lambda_R r_{\text{global}}$, combining the mean of local rewards $\bar{r}_{\text{local}}$ with the global reward $r_{\text{global}}$ using hyperparameter $\lambda_R$. Likewise, for multi-agent PPO, blended local rewards per agent $i$ may be obtained as $(1 - \lambda_R)r_i + \lambda_R r_{\text{global}}$. Setting $\lambda_R = 1.0$ provides purely global rewards (used by default by our PPO implementation), setting $\lambda_R = 0.0$ provides local rewards (used by default by our Multi-Agent PPO (MAPPO) implementation), and for experiments with mixed rewards we use $\lambda_R = 0.5$. We provide results for non-default reward configuration in Appendix C.5.

# B  Hyperparameters

This section lists the hyperparameter values used by default in our experiments. Neural network modules and training algorithms are implemented in PyTorch and PyTorch Geometric (Fey & Lenssen, 2019).

```
typedef struct ObservedGlobal
{
    long int observerId;  // -1 for global

    uint64_t tx;
    uint64_t reTx;
    uint64_t inFlight;
    uint64_t rx;
    uint64_t delivered;
    uint64_t ttlDropped;
    uint64_t tcpDiscarded;
    uint64_t givenUp;  // dropped + TCP-discarded + lost

    double maxDelay;
    uint32_t delayCount;  // used to calculate ep.-wise global avg.
    double avgDelay;
    uint32_t jitterCount;  // used to calculate ep.-wise global avg.
    double avgJitter;
    uint32_t tcpSenderSrttCount;  // used to calculate ep.-wise global avg.
    double avgTcpSenderSrtt;

    uint32_t udpRemainingFlows;
    uint64_t udpRemainingFlowBytes;
    uint64_t udpRemainingFlowDatarateSum;
    uint32_t tcpRemainingFlows;
    uint64_t tcpRemainingFlowBytes;
    uint64_t tcpRemainingFlowCWndSum;

    uint64_t txQueueCapacity;  // total capacity of all tx queues (bytes)
    uint64_t txQueueLoad;  // total load of all tx queues (bytes)

    uint64_t transmitted;  // incl. packet headers and overhead
    uint64_t transmittable;  // max. bytes sendable in cur. time frame
    uint64_t macTxDropped;
    uint64_t phyRxDropped;

    double dropRatio;  // given-up pkts. / given-up + delivered pkts.
    double inFlightRatio;  // in-flight pkts. / throughput
    double maxLinkUtilization;  // max. LU over all links
    double avgLinkUtilization;  // avg. LU over all links
    double maxTxQueueRelativeLoad;  // over all links
    double avgTxQueueRelativeLoad;  // over all links

} ObservedGlobal;

typedef struct ObservedEdge
{
    long int observerId;  // -1 for global/non-delay
    uint32_t src;
    uint32_t dst;
    bool isLinkUp;

    uint64_t channelDataRate;
    uint32_t channelDelay;
    uint32_t txQueueCapacity;
    uint32_t txQueueLoad;

    uint64_t transmitted;
    uint64_t transmittable;
    uint64_t macTxDropped;
    uint64_t phyRxDropped;

    double linkUtilization;
    double txQueueRelativeLoad;

} ObservedEdge;
```

```
typedef struct ObservedNode
{
    long int observerId;  // -1 for global
    uint32_t nodeId;

    uint64_t txAsSender;
    uint64_t txAsReceiver;
    uint64_t reTxAsSender;
    uint64_t reTxAsReceiver;
    uint64_t inFlightAsSender;
    uint64_t inFlightAsReceiver;
    uint64_t rxAsSender;
    uint64_t rxAsReceiver;
    uint64_t deliveredAsSender;
    uint64_t deliveredAsReceiver;
    uint64_t ttlDroppedAsSender;
    uint64_t ttlDroppedAsReceiver;
    uint64_t ttlDroppedAsDropper;
    uint64_t tcpDiscardedAsSender;
    uint64_t tcpDiscardedAsReceiver;
    uint64_t givenUpAsSender;  // incl. dropped and TCP-discarded
    uint64_t givenUpAsReceiver;  // incl. dropped and TCP-discarded
    uint64_t givenUpAsDropper;  // incl. dropped and TCP-discarded

    double maxDelayAsSender;
    double maxDelayAsReceiver;
    double avgDelayAsSender;
    double avgDelayAsReceiver;
    double avgJitterAsSender;
    double avgJitterAsReceiver;
    double avgTcpSenderSrttAsSender;
    double avgTcpSenderSrttAsReceiver;

    uint32_t udpRemainingFlowsAsSender;
    uint32_t udpRemainingFlowsAsReceiver;
    uint64_t udpRemainingFlowBytesAsSender;
    uint64_t udpRemainingFlowBytesAsReceiver;
    uint64_t udpRemainingFlowDatarateSumAsSender;
    uint64_t udpRemainingFlowDatarateSumAsReceiver;
    uint32_t tcpRemainingFlowsAsSender;
    uint32_t tcpRemainingFlowsAsReceiver;
    uint64_t tcpRemainingFlowBytesAsSender;
    uint64_t tcpRemainingFlowBytesAsReceiver;
    uint64_t tcpRemainingFlowCWndSumAsSender;
    uint64_t tcpRemainingFlowCWndSumAsReceiver;

    uint64_t transmittedAsEdgeSrc;  // incl. all headers and overhead
    uint64_t transmittedAsEdgeDst;  // incl. all headers and overhead
    uint64_t transmittableAsEdgeSrc;  // max. bytes sendable in cur. time frame
    uint64_t transmittableAsEdgeDst;  // max. bytes sendable in cur. time frame
    uint64_t macTxDroppedAsEdgeSrc;
    uint64_t macTxDroppedAsEdgeDst;
    uint64_t phyRxDroppedAsEdgeSrc;
    uint64_t phyRxDroppedAsEdgeDst;

    double dropRatioAsSender;  // given-up pkts. / given-up + delivered pkts.
    double dropRatioAsReceiver;  // given-up pkts. / given-up + delivered pkts.
    double inFlightRatioAsSender;  // in-flight pkts. / throughput
    double inFlightRatioAsReceiver;  // in-flight pkts. / throughput

    double maxLinkUtilizationAsEdgeSrc;  // over all links starting at this node
    double maxLinkUtilizationAsEdgeDst;  // over all links leading to this node
    double avgLinkUtilizationAsEdgeSrc;  // over all links starting at this node
    double avgLinkUtilizationAsEdgeDst;  // over all links leading to this node
    double maxTxQueueRelativeLoadAsEdgeSrc;  // over all links starting at this node
    double maxTxQueueRelativeLoadAsEdgeDst;  // over all links leading to this node
    double avgTxQueueRelativeLoadAsEdgeSrc;  // over all links starting at this node
    double avgTxQueueRelativeLoadAsEdgeDst;  // over all links leading to this node

} ObservedNode;
```

Figure 10: Our implementation of observed node, edge and global attribute vectors.

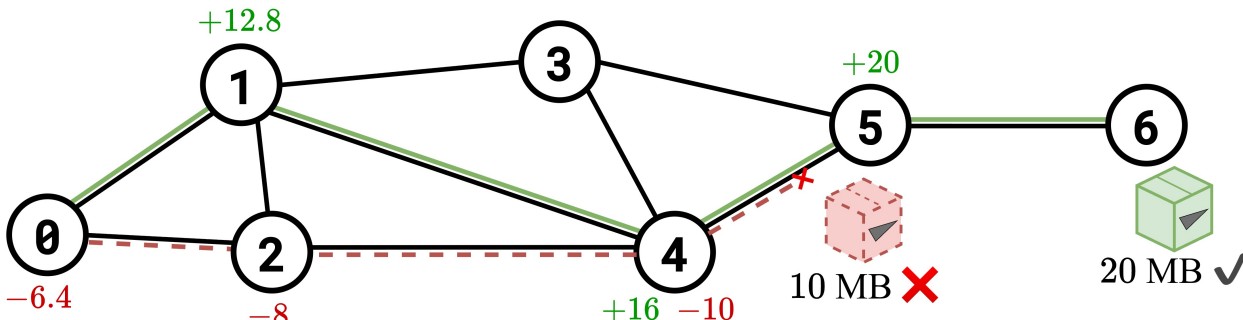

Figure 11: Illustration of local reward assignment for two packets sent from node 0 to node 6. The green/solid packet reaches its destination, yielding a reward to the three previous nodes on its path that spatially decays with $\lambda_{\text{decay}} = 0.8$. Likewise, the red/dashed packet is dropped at or before node 5 and yields a penalty to the three previous nodes on its path.

## B.1 *LOGGIA* architecture

For the Message Passing Network (MPN) stage of our algorithm *LOGGIA*, which is also denoted as a function $\zeta_\theta$, we use $L = 4$ message passing steps and use Multilayer Perceptrons (MLPs) for the global-, node-, and edge feature processing functions $f_U$, $f_V$, $f_E$, with identical configuration as shown in Table 1. For the permutation-invariant aggregation function $\bigoplus$, we use a concatenation of mean and minimum. Additionally, we apply layer normalization to each $f_U$, $f_V$, $f_E$ and wrap them in residual connections. Finally, the MPN architecture of *LOGGIA* stores the previous output link weights $\mathbf{w}_{t-1}$ (zero-initialized) and supplies them as additional feature for the next input graph.

Table 1: Hyperparameters for the feature processing MLPs $f_U$, $f_V$, $f_E$ of *LOGGIA*'s MPN stage $\zeta_\theta$.

| Hyperparameter | Symbol | Value |
|---|---|---|
| Hidden layers in MLPs $f$ | | 2 |
| MLP hidden dimension | $d_\zeta$ | 32 |
| Activation function | | LeakyReLU |
| Dropout | | 0 |

## B.2 Training Algorithms

We use PPO with Generalized Advantage Estimation (GAE) (Andrychowicz et al., 2020), separate parameters $\theta$, $\phi$, $\rho$, for policy $\pi_\theta$, value function $W_\phi$, and exploration regulator $\alpha$ with separate Adam optimizers (Kingma & Ba, 2014). The value function $W_\phi$ is comprised of an MPN identical to $\pi_\theta$, followed by a node readout MLP configured as per Table 1 and a global max pooling operation to obtain a graph-level value estimate. Hyperparameters are listed in Table 2. For our interactive Imitation Learning (IL) algorithm, we use a learning rate of $\lambda_{\text{IL}} = 0.5e\text{-}4$. Both PPO and IL training normalize observations akin to Schulman et al. (2017).

Table 2: Hyperparameters for our PPO algorithm.

| Hyperparameter | Symbol | Value |
|---|---|---|
| Policy learning rate | $\lambda_\pi$ | 3e-4 |
| Value function learning rate | $\lambda_W$ | 1e-3 |
| Exploration regulator learning rate | $\lambda_\rho$ | 3e-4 |
| Discount factor | $\gamma$ | 0.95 |
| Policy clip range | $\epsilon_\pi$ | 0.5 |
| Value function clip range | $\epsilon_W$ | 0.3 |
| Minibatches per epoch | | 16 |
| Update epochs | | 10 |
| Policy gradient clip norm | | 0.5 |
| GAE weighting parameter | $\lambda_{\mathrm{GAE}}$ | 0.9 |
| Maximum KL (for $\pi_\theta$ early stopping) | $\delta_{\mathrm{KL}}$ | 10 |
| Maximum clip fraction (for $\pi_\theta$ early stopping) | $\delta_{\mathrm{clip}}$ | 0.2 |
| Target entropy (per edge) | $\mathcal{H}_{\mathrm{targ}}$ | 0.2 |

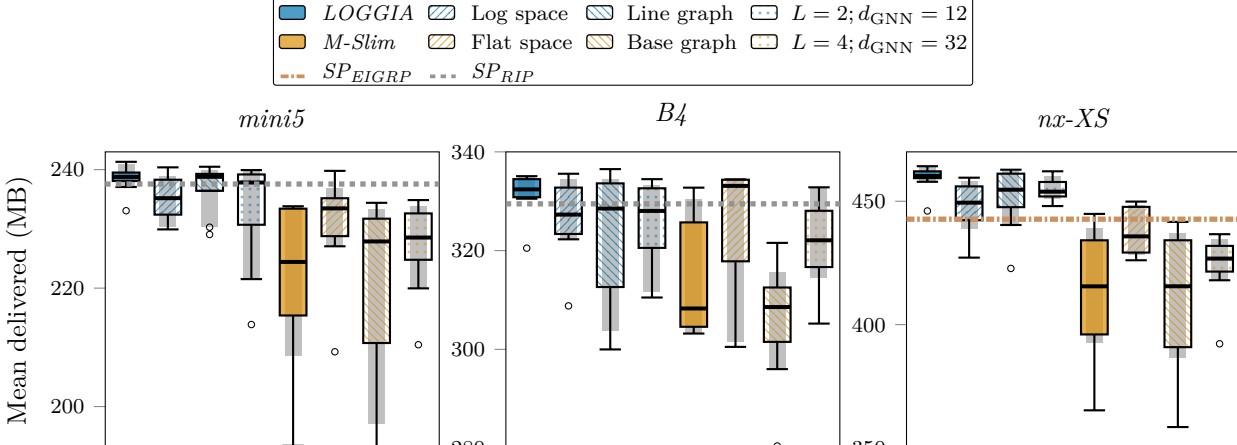

Figure 12: Architecture ablation on *LOGGIA* for various topology presets, evaluated in delay-aware deployment. Dashed lines denote the best *SP* baseline per evaluation. Semi-transparent bars overlaid on the boxplots represent the 95% bootstrapped confidence intervals. Each of the architectural choices matter in *LOGGIA*'s overall design. For *M-Slim*, predicting link weights in logarithmic space or increasing neural network size improves its performance, while there is no clear benefit from operating on the base graph instead of the line digraph.

## C  Parameter Study and Analysis

This section includes ablation studies on architectural and algorithmic aspects of *LOGGIA* and our training protocol. Unless mentioned otherwise, we use the hyperparameters noted in Appendix B.

### C.1  Architectural Details

We ablate three policy design options that distinguish *LOGGIA* from *M-Slim*: **(i)** predicting link weights in logarithmic space, **(ii)** operating on the input graph directly, instead of converting it into a line digraph, and **(iii)** using $L = 4$ MPN steps and a latent dimension $d_{\mathrm{GNN}} = 32$ instead of *M-Slim*'s $L = 2$ and $d_{\mathrm{GNN}} = 12$.

Figure 12 shows results for *LOGGIA* when disabling the aforementioned options one-by-one, and for *M-Slim* when enabling the aforementioned options one-by-one. Overall, we find that each of the adjustments play

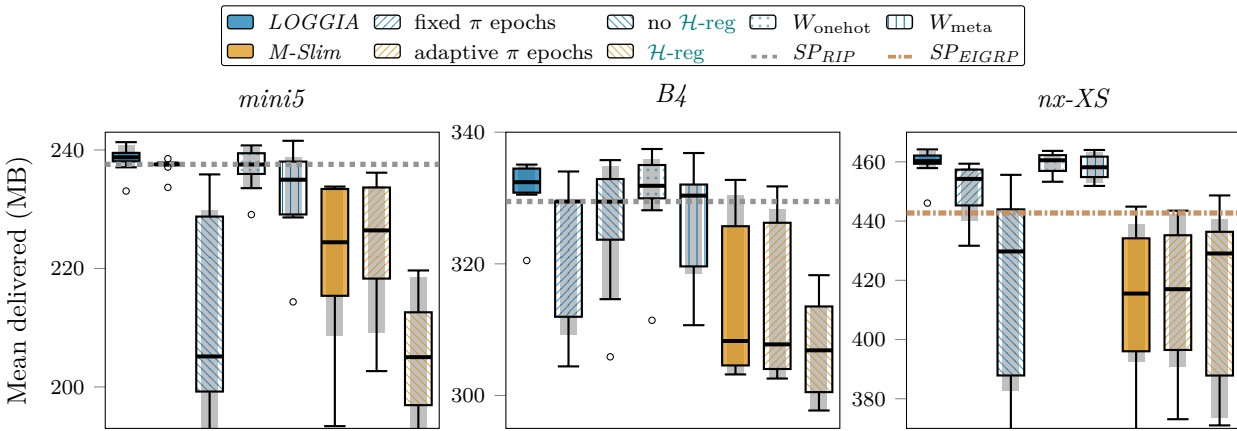

Figure 13: Policy training ablation on *LOGGIA* for various topology presets, evaluated in delay-aware deployment. Dashed lines denote the best *SP* baseline per evaluation. Semi-transparent bars overlaid on the boxplots represent the 95% bootstrapped confidence intervals. Our policy training improvements help *LOGGIA* attain higher routing performance, while their effects on *M-Slim* are inconclusive. Adding principled value learning mechanisms does not improve *LOGGIA*'s performance, indicating that value learning is not the main performance bottleneck in the evaluated scenarios.

their role in *LOGGIA*'s performance, and their combination shows the lowest variance in performance. On the other hand, *M-Slim* benefits from log-space prediction and a bigger policy network but not so clearly from skipping the line digraph conversion.

## C.2   Policy and Value Function Training

**Policy Updates.** We ablate the following two policy training adjustments: **(i)** stopping policy updates early if the estimated mean KL divergence or policy clip fraction exceed predefined thresholds $\delta_{\mathrm{KL}}$ or $\delta_{\mathrm{clip}}$ (denoted as "adaptive $\pi$ epochs" in Figure 13), **(ii)** incorporating adaptive-entropy exploration inspired by Soft Actor-Critic (SAC) with an adaptive learnable entropy scaling (denoted as $\mathcal{H}$-reg).

**Input-Dependent Value Functions.** For input-driven MDPs, Mao et al. (2018) argues that training input-dependent baselines (e.g., value functions) helps to distinguish policy-induced from exterior effects on the observed reward. This ought to reduce policy gradient variance during training and thus yielding better policies. Therefore, for *LOGGIA* only, we attempted providing principled information to its value function during training: **(i)** We one-hot encode the current event sequence number as well as the current timestep (denoted as $W_{\mathrm{onehot}}$). This requires generating at least two different event sequences per topology in the training scenario set, as well as using the same network scenarios for all training iterations, in order to render the one-hot encoding meaningful. **ii)** We implement the value function meta-learning approach of Mao et al. (2018), which uses Model-Agnostic Meta Learning (MAML) (Finn et al., 2017) to update the "main" value function from adapted baselines provided by value sub-functions (denoted as $W_{\mathrm{meta}}$). This requires repeating every input sequence seen during training at least once in a second training episode, in order to enable splitting the training trajectories into two buckets for the two value sub-function adaptations.

Figure 13 shows results for *LOGGIA* when disabling the above policy training options one-by-one, and for *M-Slim* when enabling them one-by-one. Additionally, we show results for *LOGGIA* with either of the above value learning ablations. We see that *LOGGIA* benefits from both the early stopping for policy updates and the entropy-regularized exploration, while the effects of these improvements on *M-Slim*'s performance are inconclusive. We also note that principled value learning does not improve final routing performance nor reduce performance variance, indicating that value learning is not the main performance bottleneck in the evaluated scenarios.

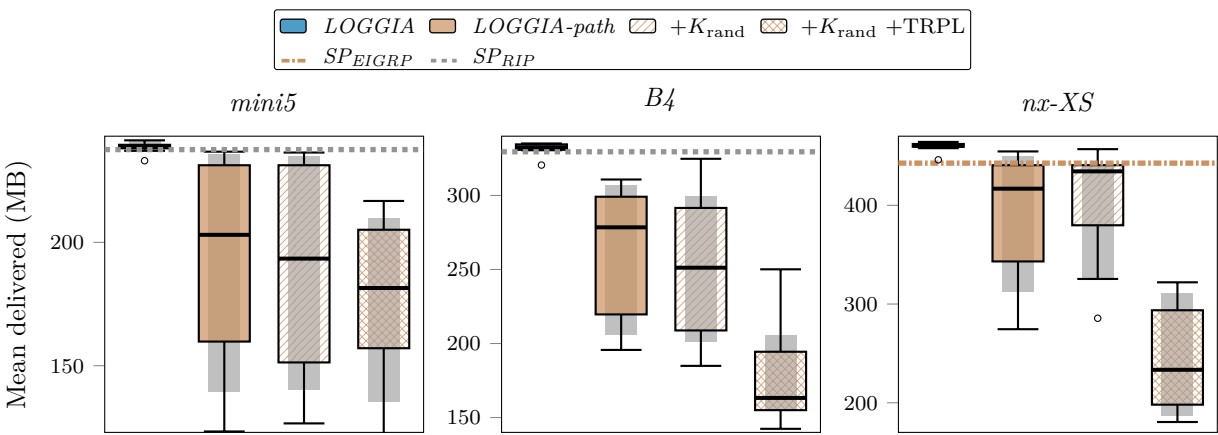

Figure 14: Path-level exploration ablation on *LOGGIA* for various topology presets, evaluated in delay-aware deployment. Dashed lines denote the best *SP* baseline per evaluation. Semi-transparent bars overlaid on the boxplots represent the 95% bootstrapped confidence intervals. *LOGGIA-path* shows sharply degrading routing performance, including the variants with added random paths or trust-region regularization. This indicates that *LOGGIA*'s edge-level exploration may be sufficient for our routing problem despite exploration happening on a purely local basis.

## C.3 Path-Level Exploration

By default, *LOGGIA* uses edge-level exploration during Reinforcement Learning (RL) training by sampling link weights from a Log-Normal distribution constructed from its per-edge output values $(\mu, \sigma)$. Yet, the complex dependency between link weights and shortest paths render the question whether higher-level exploration mechanisms like path sampling further accelerate or stabilize training. We thus create a single-agent centralized variant *LOGGIA-path* that implements exploration by sampling from a set of candidate paths per pair of source and destination node. *LOGGIA-path* uses the per-edge $\exp(\mu)$ and Yen's algorithm (Yen, 1970) to compute up to $K_{\text{top}}$ shortest paths per node pair (at least one due to our assumption of graph connectedness). For each node pair, the normalized path costs $\tilde{c}_i = c_i - \min_{j \in [0, K-1]} c_j$ then form logits for a categorical distribution over the corresponding paths $p_i$ from which a path is sampled. All sampled paths together are then converted to routing actions analog to *LOGGIA* in single-agent deployment. Note that this turns *LOGGIA*'s continuous action space into a discrete path selection space. We set $K_{\text{top}} = 5$ for the following ablation and consider two variants for *LOGGIA-path*: **(i)** For each pair of source and destination node, we add up to $K_{\text{rand}} = 3$ random paths to the set of $K \leq K_{\text{top}}$ least-cost paths by computing the shortest path on perturbed weights $e^{\mu + \iota}$, with $\iota \sim \mathcal{N}(0, 1)$, and adding it to the existing path candidate set if it is distinct from the existing candidates. This variant is denoted as "$+K_{\text{rand}}$" in Figure 14. **(ii)** We adopt discrete trust-region regularization (Otto et al., 2020; Becker et al., 2025) with target KL 0.5 and projection loss coefficient $\alpha_{\text{TRPL}} = 0.1$ for *LOGGIA-path*'s now discrete action space (denoted as +TRPL in Figure 14).

The results shown in Figure 14 indicate that path-level exploration does not improve routing performance, neither for smaller network topologies like *M5* with fewer candidate paths per node pair nor for the larger *B4* topology. In fact, routing performance degrades sharpy with path-level exploration, including the variants with added random paths or trust-region regularization. This indicates that *LOGGIA*'s edge-level exploration may be sufficient for our routing problem despite exploration happening on a purely local basis.

## C.4 Imitation Learning Pretraining

As an alternative to interactive *DAgger*-style IL (Ross et al., 2011), we can choose to forgo the interaction loop and generate a dataset that contains tuples of network topology with just the initial observation of an episode (i.e., a computer network with zero utilization), paired with student and expert target action. We denote this offline IL approach, using the supervised loss function $\mathcal{L}_{\text{IL}}$ introduced in Section 4.3, as $\text{IL}_{\text{Initial}}$, as opposed to $\text{IL}_{\text{DAgger}}$ for DAgger-style IL. We investigate the performance of *LOGGIA* trained

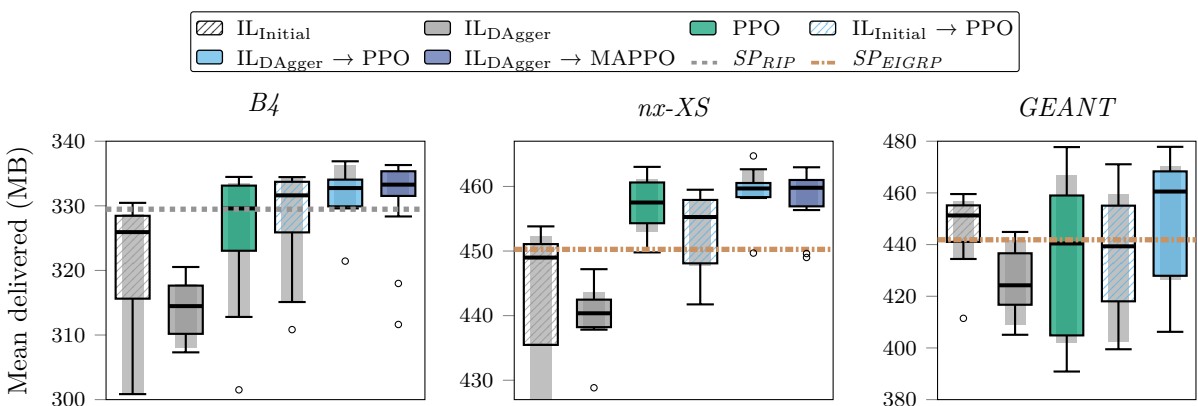

Figure 15: Performance of *LOGGIA* for varying topology presets and training algorithm combinations, evaluated in `Local-Multi` deployment. Dashed lines denote the best *SP* baseline per evaluated topology preset. Semi-transparent bars overlaid on the boxplots represent the 95% bootstrapped confidence intervals. IL$_{\mathrm{DAgger}}$ is inferior to PPO as a standalone trainer, but improves subsequent PPO training consistently. While IL$_{\mathrm{Initial}}$ outperforms IL$_{\mathrm{DAgger}}$ as a standalone trainer, it is inferior to IL$_{\mathrm{DAgger}}$ as a pretraining phase.

with our PPO/MAPPO and IL implementations individually, and PPO/MAPPO in conjunction with IL pretraining. For the IL$_{\mathrm{Initial}}$ training phase, we run 20 training iterations with 10 times the amount of scenarios used for PPO/IL$_{\mathrm{DAgger}}$, to account for the reduced number of data points due to not interacting with the environment. All training is done using delay-aware observers.

The results of Figure 15 show that, on its own, *DAgger*-style IL is not enough to train *LOGGIA* produce competitive routing algorithms. Possibly, this is due to IL training the student to mimic a static routing algorithm, ignoring useful telemetry features. However, precluding an IL to a single- or multi-agent PPO training phase improves the delivery rate and decreases inter-seed performance variance, potentially due to "warm-starting" *LOGGIA*'s policy with training signals from expert data. Figure 15 further shows that tealIL$_{\mathrm{Initial}}$ outperforms IL$_{\mathrm{DAgger}}$ when used as a standalone training algorithm. Yet, when used as pretraining phase for PPO, tealIL$_{\mathrm{Initial}}$ is inferior to interactive IL$_{\mathrm{DAgger}}$.

## C.5 Multi-Agent PPO Variants

As an alternative to our MAPPO implementation, we implement per-agent value functions that align with Independent PPO (IPPO) (de Witt et al., 2020). Additionally, we consider a per-agent value function variant that concatenates the global/central network state to the agent-specific observation. This is congruent to the *AS* value function input ablation of Yu et al. (2022), and we denote this variant as Hybrid-Value PPO (HVPPO). For all multi-agent settings, we consider a linear combination of the global reward used by our single-agent PPO implementation, and the local rewards used by MAPPO by default (c.f. Section 4.2), subject to a weighting parameter $\lambda_R$. Reward function implementation details can be found in Appendix A.3.

The combination of value function location, observer placement, and reward mixing yields several possible combinations for multi-agent PPO training. Figure 16 shows that *LOGGIA*'s evaluation performance is very similar across most of these configurations, giving a slight edge only to the choice to include some portion of spatial reward into the per-agent rewards ($\lambda_R \neq 1$). Notably, in line with results reported in related work (Geng et al., 2025), choosing between a centralized (MAPPO), an independent (IPPO) or a hybrid (HVPPO) algorithm does not appear to matter, and we default to MAPPO for simplicity. Regarding observer placement during training, for the MAPPO algorithm with local rewards, a central observer deployment produces marginally better results than using local observers (denoted by (L) in Figure 16).

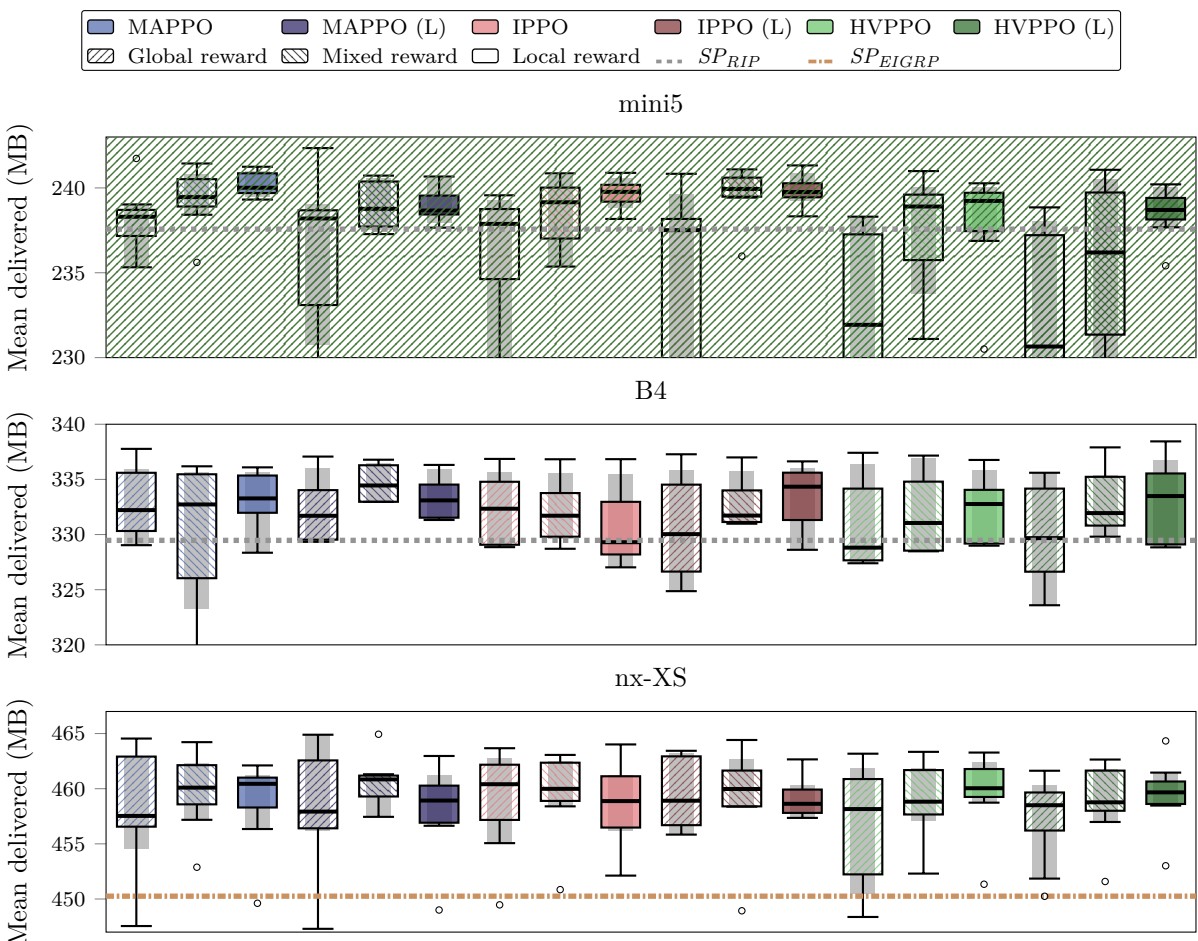

Figure 16: Performance of *LOGGIA* for various topology presets, multi-agent PPO algorithms, central vs. local training observers (the latter marked by (L)), and reward settings. All approaches are trained in the delay-aware setting with IL pretraining and evaluated in `Local-Multi` deployment. Dashed lines denote the best *SP* baseline per evaluation. Semi-transparent bars overlaid on the boxplots represent the 95% bootstrapped confidence intervals. All multi-agent algorithms show similar performance, both with centralized and local training observer deployment. For *mini5* and the *nx-XS* family, either mixed ($\lambda_R = 0.5$) or purely local ($\lambda_R = 0$) rewards marginally improve performance compared to sharing a global reward ($\lambda_R = 1$).

Table 3: Performance of *LOGGIA* and baseline algorithms across different network performance metrics, trained and evaluated on the *B4* topology. For *LOGGIA* and the neural baselines, table cells show the median over 8 random seeds; the optimization objective is delivery per episode. *LOGGIA* scores a significantly higher delivery rate than all other approaches, while maintaining competitive values for packet delay (latency) and the average packet buffer load. All neural routing algorithms show notably higher TCP-side packet discards, which in our simulation happens due to out-of-order packet arrival.

| | **Delivered** (MB) ↑ | **Delay** (ms) ↓ | **QueueLoad** (%) ↓ | **TCP Discard** (MB) ↓ |
|---|---|---|---|---|
| *SP_{RIP}* | 329.473 | 11.345 | 13.961 | **1.426** |
| *MAGNNETO* | 311.565 | 11.328 | 12.824 | 5.282 |
| *FieldLines* | 296.184 | 12.163 | 14.471 | 3.520 |
| *M-Slim* | 310.988 | **11.178** | **12.440** | 5.426 |
| *LOGGIA* (ours) | **333.370** | 11.424 | 14.091 | 9.561 |

Table 4: Performance of *LOGGIA* and baseline algorithms across different network performance metrics, trained and evaluated on the *GEANT* topology. For *LOGGIA* and the neural baselines, table cells show the median over 8 random seeds; the optimization objective is delivery per episode. *LOGGIA* scores a significantly higher delivery rate than all other approaches, and outperforms other approaches in terms of packet delay (latency) and the average packet buffer load. All neural routing algorithms show notably higher TCP-side packet discards, which in our simulation happens due to out-of-order packet arrival.

|  | **Delivered** (MB) ↑ | **Delay** (ms) ↓ | **QueueLoad** (%) ↓ | **TCP Discard** (MB) ↓ |
|---|---|---|---|---|
| $SP_{EIGRP}$ | 441.861 | 9.375 | 16.361 | **3.184** |
| *MAGNNETO* | 420.941 | 9.662 | 15.517 | 4.875 |
| *FieldLines* | 450.328 | 9.418 | 16.208 | 3.711 |
| *M-Slim* | 438.191 | 9.750 | 16.629 | 7.616 |
| *LOGGIA* (ours) | **460.714** | **9.002** | **14.696** | 8.725 |

# D    Additional Results

## D.1    Multi-Metric Performance

Tables 3 and 4 provide supplementary results for the experiment of Section 6.1, showing the performance of the evaluated algorithm with respect to other metrics besides the (episodic) delivery, which is our optimization objective. We find that our algorithm *LOGGIA* achieves competitive performance also with respect to average packet delay (latency) and average packet buffer load, although *M-Slim* slightly outperforms *LOGGIA* on *B4* with respect to these two metrics. On the other hand, we observe that all neural routing algorithms provoke a notably higher number of packets dropped by the receiving TCP socket, which in our simulation happens if packets arrive out-of-order. It is conceivable that, in some situations, the neural routing algorithms provoke path changes with big differences in total path latency, leading to TCP packets arriving out-of-order and thus to retransmissions. Future work may improve on this by training more nuanced route calculation policies specifically for TCP flows, e.g., based on the expected volume of re-routed traffic (Ye et al., 2022).

## D.2    The Influence of Control Granularity and Buffer Size

For the *B4* topology, we evaluate different network environment settings with reduced (50%) or increased (200%) packet buffer size, and a coarser ($H = 200$, $\tau = 10$ ms) or finer ($H = 1000$, $\tau = 2$ ms) granularity of routing control. The results are shown in Figure 17. We find that *LOGGIA*'s performance decreases with smaller packet buffer size, but does not increase consistently with larger buffers. Further, for *M-Slim* and *LOGGIA*, the delivery rate increases slightly with a more fine-grained control loop (albeit at the expense of a larger variance in performance), whereas *MAGNNETO* appears to work best with a coarser control loop.

## D.3    Communication and Inference Delays in the *GEANT* topology

Section 6.2 investigates the effect of respecting communication/inference delays during training on the *B4* topology. Here, we repeat this experiment on the *GEANT* topology. Figure 18 shows the results, which confirm our previous findings: i) Including communication and inference delays in the interaction model decreases overall performance, as seen in the comparison between `Birdseye-Single` and the other deployment modes. ii) Of all delay-aware observer and agent deployments, the `Local-Multi` mode with local observers and agents yields the best results. iii) Routing performance decreases with an increasing delay scaling factor, i.e., higher inference delays. iv) Respecting delays during shows no clear effect for single-agent training. Compared to the evaluation *B4* of Section 6.2, the relative performance decrease caused by an increasing $\lambda_{ac}$ is more severe on the larger *GEANT* topology, presumably due to the larger inference times.

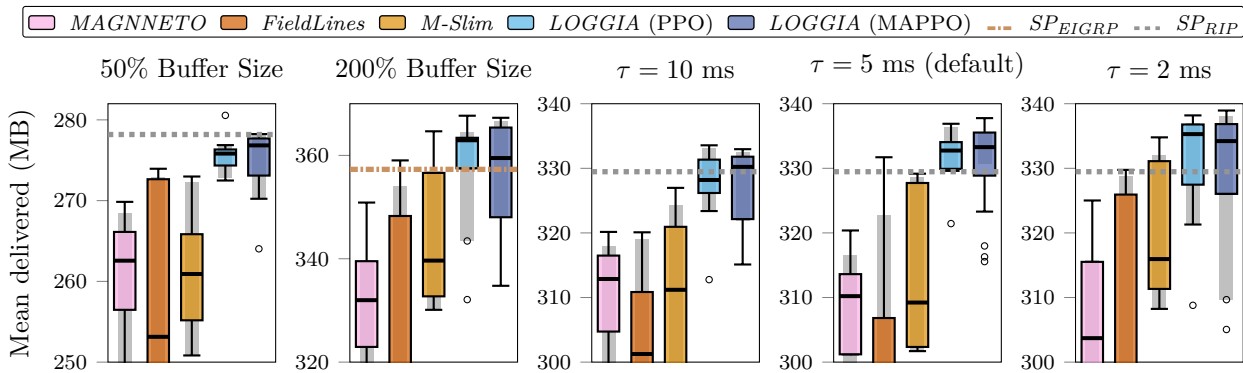

Figure 17: Performance of *LOGGIA* trained and evaluated in single-/multi-agent mode on the *B4* topology, with network setting variations as described in the diagram headers ($\tau = 5$ ms corresponds to the default setting). All approaches are trained in the delay-aware setting with IL pretraining and evaluated in `Local-Multi` deployment. Dashed lines denote the best *SP* baseline per evaluation. Semi-transparent bars overlaid on the boxplots represent the 95% bootstrapped confidence intervals. We find that *LOGGIA*'s performance decreases with smaller packet buffer size, but does not increase consistently with larger buffers. Further, for *M-Slim* and *LOGGIA*, the delivery rate increases slightly with a more fine-grained control loop (albeit at the expense of a larger variance in performance), whereas *MAGNNETO* appears to work best with a coarser control loop.

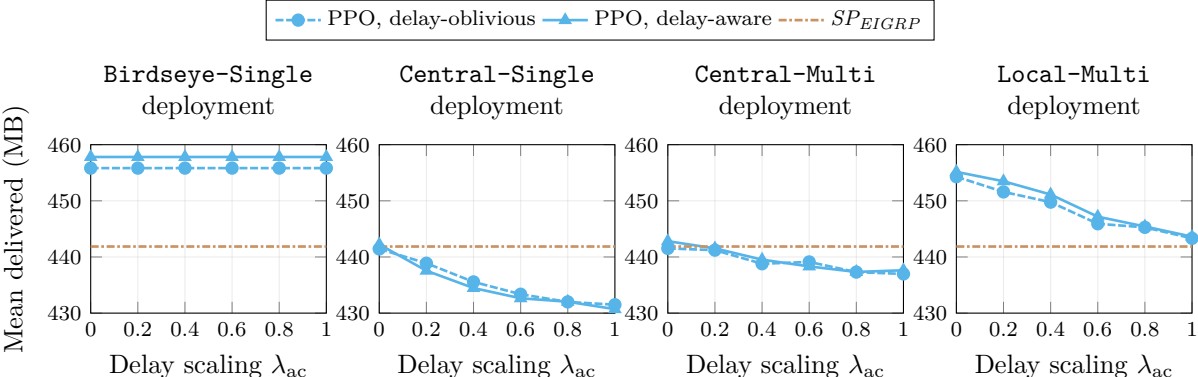

Figure 18: Performance of *LOGGIA* trained with PPO/MAPPO in delay-aware/delay-oblivious configuration, and evaluated in different deployment modes on the *GEANT* topology. Solid lines show the interquartile mean across 8 random seeds per approach. Dashed lines denote the best *SP* baseline per evaluation. All runs include IL pretraining and are evaluated for varying inference delay scalings $\lambda_{ac} \in [0, 1]$ (x-axis per plot). The findings match those of Figure 7: Including communication and inference delays in the interaction model decreases overall performance, as seen in the difference between the delay-oblivious `Birdseye-Single` mode and the other delay-aware modes. Of the delay-aware deployments, the `Local-Multi` mode with fully distributed observers and agents yields the best results. Moreover, routing performance decreases with an increasing delay scaling factor, i.e., higher inference delays. Finally, respecting delays during shows no clear effect for single-agent training.

Table 5: Mean delivery rate and inference time of *LOGGIA* over 8 random seeds, trained and evaluated on *GEANT* for two different processors. On the faster Intel Xeon Gold 6448Y, *LOGGIA* runs notably faster and delivers slightly more data in the delay-aware setting ($\lambda_{ac} = 1$), despite using the same hyperparameters.

| Processor | Inference Time (ms) | Delivered (MB) ($\lambda_{ac} = 0$) | Delivered (MB) ($\lambda_{ac} = 1$) |
|---|---|---|---|
| Xeon 6780E (default) | 7.466 | 450.972 | 440.619 |
| Xeon Gold 6448Y | 4.985 (−33.2%) | 450.878 (−0.02%) | 444.609 (+0.91%) |

### D.4 The Influence of Hardware Speed on Routing Performance

The results of Section 6.2 show that the performance of neural routing algorithms, in our delay-aware setting, depends on their inference speed. The inference speed, in turn, depends on the hardware the algorithm runs on. Table 5 shows the performance and mean inference time of *LOGGIA*, trained and evaluated on the *GEANT* topology, for two different processors: The Intel Xeon 6780E is our default option and provides a turbo clock speed of 3.0 GHz. The Intel Xeon Gold 6448Y, on the other hand, provides a higher turbo clock speed of 4.1 GHz. On average, *LOGGIA*'s inference times are notably lower on the faster Xeon Gold 6448Y, and thus in the inference-delay-aware setting ($\lambda_{ac} = 1$) *LOGGIA* performs slightly better despite equal training and evaluation hyperparameters. When disregarding inference delays during training and evaluation ($\lambda_{ac} = 0$), the reported performance is comparable across both processors.

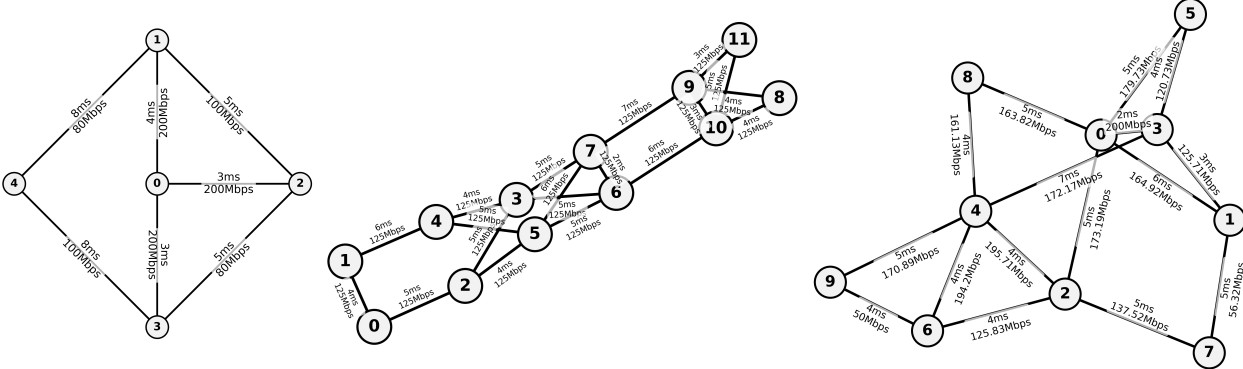

Figure 19: Visualizations of the *mini5*, *B4* and an example *nx-XS* topology with link datarates and delays.

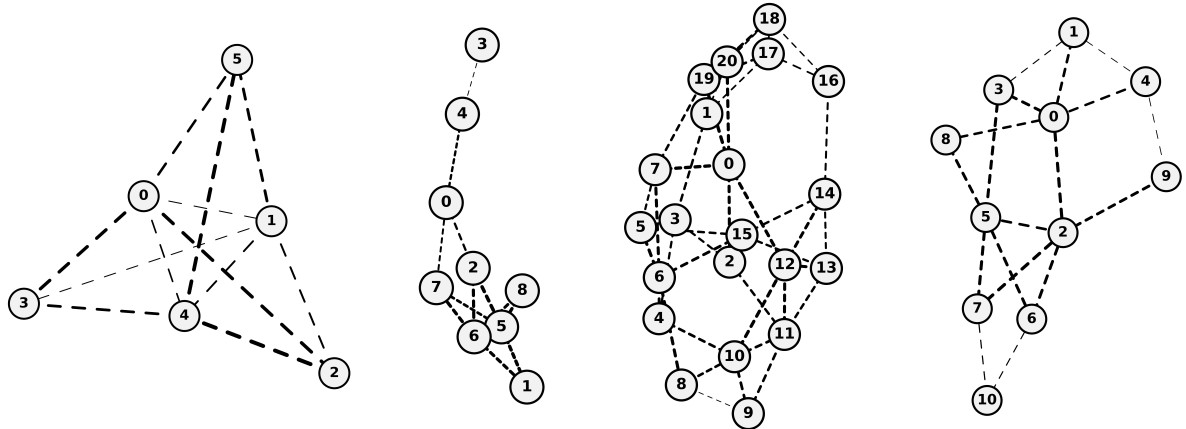

Figure 20: Additional visualizations of two topologies of the *nx-XS* family (Left), and of the *nx-S* family (Right). Thicker edges denote larger link datarate, fewer dashes denote lower link latency.

