# OpenReview forum: "Towards Near-Real-Time Telemetry-Aware Routing with Neural Routing Algorithms"
_TMLR — Under review for TMLR_

### Review · Reviewer_u5xj · 2026-05-17

**Summary Of Contributions:**

This paper tackles the problem of telemetry-aware routing in computer networks under realistic communication and inference delays. The authors make two main contributions: (1) a packet-level simulation framework that models delay-aware state aggregation and action dissemination, and (2) a novel neural routing algorithm named LOGGIA, which combines a GNN-based log-space link weight predictor with imitation learning pretraining and on-policy RL (PPO).

**Audience:**

Yes

**Audience Explanation:**

The paper focuses on an interesting topic of graph data mining via GNN models.

**Claims And Evidence:**

Yes

**Claims Explanation:**

The paper provides a detailed experimental validation across multiple datasets.

**Requested Changes:**

1.In section 4.1 $x_e$ and $x_u$ adopt the same learning function (implemented by different MLPs), while $x_v$ adopts a different one. However, the authors do not introduce the motivation for this design. Could you provide some insights for this design?

2.The update of the $x_e$, $x_u$ and $x_v$ only involves one node pair $(v, u)$. While the MPNs always consider the information from first-order neighbors. How do you apply MPNs for node representation learning. Moreover, all formulas in the paper should be numbered.

3.Since the authors adopt a basic MPNs for node representation learning, does the choice of MPN backbone have a significant impact on performance?

---

> ### Author Response · Authors · 2026-05-29
>
> Thank you very much for your review, which provides the the opportunity to clarify our writing concerning MPNs. We have updated the paper text and the notation for MPN feature updates to answer your questions and add more clarity to the presentation (adjusted text is colored in orange in the updated version).
>
>
> > 1. In section 4.1 and adopt the same learning function […]
> >
> > 2. The update of the only involves one node pair […]
> >
>
> The edge update function $f_E$ updates the latent representation of each edge $e = (v, u)$ from its own representation $\mathbf{x}\_{e}$, the representations of the incident nodes $v$ and $u$, and the global features. In contrast, the global update function $f_U$ updates the global representation using the previously updated and aggregated representations $\bigoplus\_{v' \in V} \mathbf{x}^{l+1}\_{v'}$ *and* $\bigoplus\_{e' \in E} \mathbf{x}^{l+1}\_{e'}$ of all nodes and edges. Concerning the implementation of $f_E$ and $f_V$, the update happens in parallel for all edges/nodes using the adjacency matrix $\mathbf{A}$.
>
> > 2. […] Moreover, all formulas in the paper should be numbered.
>
> Thank you for pointing this out. We have now numbered all our equations.
>
> > 3. […] does the choice of MPN backbone have a significant impact on performance?
>
> Bronstein et al. [1] define three “flavors” for graph neural networks, which cover most of the existing GNN designs with first-order spatial aggregation/propagation. Of these, we decided to use MPNs, as they are a member of the most general flavor in terms of representational capacity, and additionally support processing of edge and global feature. While we have not tried other variants, our algorithm reliably reaches convergence during training, indicating that our network architecture is suitable for the given problem.
>
> [1]: Bronstein, Michael M., et al. "Geometric deep learning: Grids, groups, graphs, geodesics, and gauges." *arXiv preprint arXiv:2104.13478* (2021).

---

### Review · Reviewer_4ekA · 2026-05-18

**Summary Of Contributions:**

This work proposes a framework as well as several schemes to perform network routing under different assumptions (e.g. whether only local information, or full network information is available for a node's routing decisions).

**Audience:**

Yes

**Audience Explanation:**

Simulating and solving routing problems (network or otherwise) are common usecases for RL and the environment implemented here seems to be well thought out.

**Broader Impact Concerns:**

I do not see broader impact concerns

**Claims And Evidence:**

Yes

**Claims Explanation:**

Since my background is in Reinforcement Learning, my critique will mostly cover the problem from an RL POV while assuming the actual network simulation and other networking specifics are sound.
For networking specific concerns, please refer to other reviewers which may have a deeper background than I have.

The evaluation is done under different network topologies  and latencies under different traffic patterns and seems to be sound in the grand scheme of things.

One thing missing is a measurement of run-to-run variance which can be more important than final performance (an algorithm which has high run-to-run variance may not be deployable even if peak performance may be higher).
We recommend the authors have a look at the RL best practices outlined in https://arxiv.org/abs/2108.13264 (i.e. at least N=10 independent runs, median/interquartile mean as a robust estimator of mean, bootstrap CI,...)

More importantly, the paper does not make clear what components are prior contributions and what is actually novel.
For instance this paper makes the statement

> Our own approach LOGGIA introduces a log-space link-weight parameterization together with RL training stabilizers inspired by maximum-entropy methods

However, this is not special: Entropy bonuses are standard in all policy-based algorithms (see e.g. the famous "37 implementation details" published at ICLR 2022 https://iclr-blog-track.github.io/2022/03/25/ppo-implementation-details/)
Standard libraries such as CleanRL (https://github.com/vwxyzjn/cleanrl/blob/master/cleanrl/ppo.py) or stable baselines (https://stable-baselines3.readthedocs.io/en/master/modules/ppo.html) also implement entropy coefficients.
The same thing is true with other components of the algorithm as well:
Early stopping based on KL divergences is equally standard and implemented in e.g. CleanRL as well as being part of the ablations in https://iclr-blog-track.github.io/2022/03/25/ppo-implementation-details/
This makes sense since PPO performs a quadratic approximation of TRPO's exact KL bound, which can become loose, so adding in a "backstop" based on the actual value of KL is often useful.

Even warm-starts using IL are by now pretty standard: You just don't see them in e.g. the PPO paper since that benchmarks the specific algorithm, but if you look at applications (e.g. robotics) IL warm starts are standard.
I also think there are plenty of of RL applications which are substantially related that should be mentioned, for example RL for the vehicle routing problem or the multicommodity flow problem (i.e. every source-destination pair is a commodity).
I would expect especially the latter to induce quite similar algorithms to this (-> GNN+RL)

Further, I do not really understand what you mean with the "empty" observations for behavioral cloning in appendix C4: Behavioral Cloning is essentially just supervised learning on the expert policy. The core difference between BC and DAGGER is where the dataset of expert demonstrations comes from: DAGGER performs rollouts using the policy (or a mixture of policy and expert) and then performs BC between the states from the policy and what would be the expert actions. One can think of DAGGER as BC with an on-policy state distribution.
BC does (generally) not make assumption on where the state distribution comes from (which is also why it is more prone to drift).

**Requested Changes:**

See above.
Short summary
- bootstrap CIs for methods
- clarification of which parts are novel and proper citation of existing methods

---

> ### Author Response · Authors · 2026-05-29
>
> Thank you very much for your review, and the opportunity to improve the presentation of the RL-related aspects of our work. We have updated the paper text as follows (adjusted text is colored in teal in the updated submission).
>
> > One thing missing is a measurement of run-to-run variance […]
>
> We have added bootstrap confidence intervals to the boxplots of our submission, and integrated them into the description and discussion of the results.
>
> > However, this is not special: Entropy bonuses are standard […]
> >
> > [...]
> >
> > Even warm-starts using IL are by now pretty standard: [...]
>
> Our algorithm _LOGGIA_ includes log-space link weight prediction, entropy regularized exploration akin to SAC, adaptive policy updates, and imitation learning pretraining, to greatly improve performance and decrease variance. The novelty of _LOGGIA_ lies in the combination of these adjustments, specifically for the field of neural routing, to stabilize neural routing performance. Indeed, the individual aspects themselves have been used in related fields and previous work, and we have added missing citations and slightly re-phrased the contribution statement in the introduction and the intro to the method section to make this clearer.
>
> > I also think there are plenty of of RL applications which are substantially related […]
>
> We thank the reviewer for pointing us towards important related work. The comparison to multicommodity flow is valid, and we have added pointers to related work outside of routing in computer networks.
>
> > Further, I do not really understand what you mean with the "empty" observations […]
>
> We have clarified the description of DAgger-style vs. non-interactive IL in appendix C4. The latter we now call "non-interactive IL" to avoid confusion. By “empty” observations we meant the initial observation returned by the environment before network simulation starts, which is “empty” in the sense that no utilization has happened, yet. We now call these “initial” observations in the text.

---

### Review · Reviewer_sSNp · 2026-06-04

**Summary Of Contributions:**

This paper studies telemetry-aware routing in a realistic near-real-time network environment. In such settings, routing decisions must quickly respond to traffic bursts, while telemetry collection and model inference introduce delays. To address this challenge, the authors treated telemetry-aware routing as a delay-aware closed-loop control problem. They extended PackeRL with ns3-ai shared-memory communication to build a packet-level simulation framework that models communication and inference delays. The framework supports both centralized and distributed neural routing policies for training and evaluation. They further proposed LOGGIA, which uses a Graph Neural Network (GNN) to predict link weights in log-space. LOGGIA uses Imitation Learning to get started, followed by Reinforcement Learning. Experiments on synthetic and real-world topologies show that existing neural routing approaches, such as MAGNNETO, M-Slim, and FieldLines, fail to outperform shortest-path routing when realistic delays are considered. In contrast, LOGGIA continues to achieve performance gains in the fully distributed setting and generalizes from a small training topology to unseen networks with up to 100 nodes.

**Audience:**

Yes

**Audience Explanation:**

Anyone working at the intersection of machine learning, reinforcement learning, graph neural networks, and networking systems would find this highly relevant.

**Broader Impact Concerns:**

No concerns.

**Claims And Evidence:**

Yes

**Claims Explanation:**

The central claims are: (i) existing neural routing approaches lose their advantage over shortest‑path routing once realistic delays are modeled; (ii) the proposed LOGGIA, combined with the delay‑aware framework, can still outperform such baselines in a near–real‑time setting; and (iii) LOGGIA scales and generalizes to unseen topologies while retaining this advantage.

The experimental design directly supports these claims. The authors explicitly tested their algorithm against baselines in different deployment modes (Birdseye, Central, and Local) to show how delays impact performance. They provide clear charts showing that as inference delay increases, routing performance drops. Furthermore, their ablation studies systematically break down why LOGGIA works, proving that predicting link weights in log-space and using their specific training protocol are the reasons it succeeds where others fail.

**Requested Changes:**

1. Clarify and justify the delay and communication model:
* The framework assumes out‑of‑band state/action communication with unlimited bandwidth and shortest‑delay paths, and does not model forwarding information base installation delays or in‑band telemetry overhead beyond a brief discussion in Appendix A.2.
* The paper should more clearly explain how sensitive results are to these assumptions. For example, a short sensitivity study where state messages are rate‑limited, or where forwarding information update delays are added, would help bound how “realistic” the conclusions are.

2. Analyze TCP behavior and out‑of‑order drops more deeply:
* The paper acknowledges that neural routing algorithms trigger more TCP socket drops due to out‑of‑order arrivals, but this is only briefly discussed in appendix.
* Given the focus on realistic operation, at least one main‑text figure should break down the fraction of dropped bytes by cause, and show how LOGGIA compares to shortest path baselines along this dimension.

---

> ### Author Response · Authors · 2026-06-12
>
> Thank you very much for your review, and the opportunity to clarify the presentation of our communication model and to qualify the results of _LOGGIA_. We have updated the paper text as follows (adjusted text is colored in magenta in the updated submission):
>
> > The framework assumes out‑of‑band state/action communication with unlimited bandwidth […]
> >
> > [...]
> >
> > The paper should more clearly explain how sensitive results are to these assumptions. […]
>
> Both the computational cost of updating forwarding tables, and the overhead cost of transmitting state update messages, are important aspects which are abstracted away in the scope of this work. We have updated the description of our communication model in Section 3.2 to clearly discuss these assumptions, which are limitations of our model, as well as their effect on the results we have obtained in our experiments.
>
> > The paper acknowledges that neural routing algorithms trigger more TCP socket drops […]
> >
> > [...]
> >
> > Given the focus on realistic operation, at least one main‑text figure should break down the fraction of dropped bytes […]
>
> We agree that the dropping behavior of the evaluated approaches is an interesting aspect to analyze in our work. The newly added Figure 6 depicts the different drop reasons for the first experiment shown in Figure 5. Its content matches the finding of Tables 3 and 4 of Appendix D, stating that neural routing algorithms cause more TCP drops due to out-of-order arrival. The figure also shows that _LOGGIA_ maintains competitive overall drop counts for all evaluated topology settings except _B4_, as it is able to reduce the amount of packets dropped due to full packet buffers.

---

### Author Response · Authors · 2026-06-12
**General Response to Reviewers**

We thank the reviewers for their useful and insightful comments, which give us the opportunity to improve our submission’s presentation quality and clarity. We have updated the submitted paper PDF to include the requested changes, with a reviewer-specific text color that highlights our modifications:

- Reviewer u5xj (orange text): Clarify our writing regarding MPNs; numbering equations.
- Reviewer 4ekA (teal text): Bootstrapped CIs for methods; clarification of novelty in our algorithmic approach, particularly from the RL perspective.
- Reviewer sSNp (magenta text): Extended discussion on our communication model and its assumptions/limitations; drop cause analysis in paper main text.

For the post-discussion phase version of the paper, these text colors will be removed again.

We are happy to resolve any points that remain unclear.